# Ensemble Value Functions for Efficient Exploration in Multi-Agent Reinforcement Learning

## Abstract

Existing value-based algorithms for cooperative multi-agent reinforcement learning (MARL) commonly rely on random exploration, such as $\epsilon$-greedy, to explore the environment. However, such exploration is inefficient at finding effective joint actions in states that require cooperation of multiple agents. In this work, we propose ensemble value functions for multi-agent exploration (EMAX), a general framework to seamlessly extend value-based MARL algorithms with ensembles of value functions. EMAX leverages the ensemble of value functions to guide the exploration of agents, stabilises their optimisation, and makes their policies more robust to miscoordination. These benefits are achieved by using a combination of three techniques. (1) EMAX uses the uncertainty of value estimates across the ensemble in a UCB policy to guide the exploration. This exploration policy focuses on parts of the environment which require cooperation across agents and, thus, enables agents to more efficiently learn how to cooperate. (2) During the optimisation, EMAX computes target values as average value estimates across the ensemble. These targets exhibit lower variance compared to commonly applied target networks, leading to significant benefits in MARL which commonly suffers from high variance caused by the exploration and non-stationary policies of other agents. (3) During evaluation, EMAX selects actions following a majority vote across the ensemble, which reduces the likelihood of selecting sub-optimal actions. We instantiate three value-based MARL algorithms with EMAX, independent DQN, VDN and QMIX, and evaluate them in 21 tasks across four environments. Using ensembles of five value functions, EMAX improves sample efficiency and final evaluation returns of these algorithms by 60%, 47%, and 539%, respectively, averaged across 21 tasks.

## 1 Introduction

Cooperative multi-agent reinforcement learning (MARL) (Albrecht et al., 2024) jointly trains a team of agents to maximises shared cumulative rewards and has real-world applications in autonomous driving (Shalev-Shwartz et al., 2016; Zhou et al., 2021), warehouse logistics (Li et al., 2019; Krnjaic et al., 2022), and complex physics problems (Mojgani et al., 2023; Koumoutsakos & Litvinov, 2024). Existing MARL algorithms have shown good performance in a variety of cooperation tasks (Papoudakis et al., 2021), but commonly used value-based MARL algorithms still rely on random exploration processes, such as $\epsilon$-greedy (e.g. Sunehag et al. (2018); Rashid et al. (2020)). Such random exploration is inefficient in exploring the joint action space of all agents to discover cooperation in MARL. To illustrate this inefficiency, consider the following example in which two agents have to navigate a gridworld to jointly pick-up a heavy object, visualised in Figure 1. To learn to pick-up the goal object, agents need to cooperate and both select the pick-up action in a state where both agents are next to the object. However, such cooperation is highly unlikely by following random exploration.

To address this inefficiency in learning to cooperate, it is essential for agents to focus their exploration on parts of the environment which require cooperation. To guide exploration towards states and actions that require cooperation, we follow the simple intuition that such states and actions result in varying rewards depending on the actions of other agents. In our example , each agent may receive varying rewards when selecting the pick-up action in a state where both agents are next to the goal object (Figure 1 right) since their reward depends on whether the other agent also selects the pick-up action. Only if both agents select the pick-up action do they succeed and receive a positive reward, whereas if any agent does not select the pick-up action

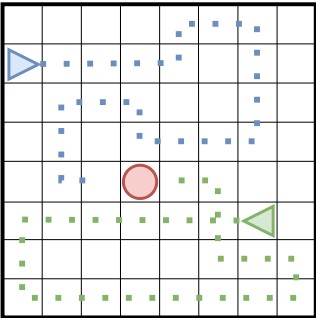 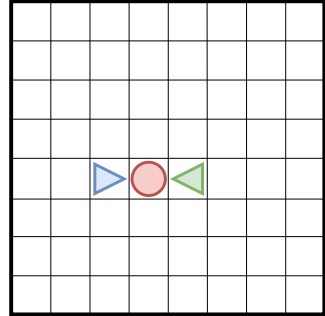

Figure 1: Motivational example: Two agents (triangles) can independently explore their movement (left), but rely on joint cooperation to pick-up the heavy circular goal object (right). To overcome the inefficiency of random exploration in discovering such cooperation, we leverage uncertainty across ensembles of value functions to guide the multi-agent exploration towards state-action pairs that require cooperation.

they do not receive such a reward. In contrast, each agent receives the same reward of zero whenever selecting an action to move independently of the actions of the other agent (Figure 1 left) because there is no potential for rewards if any one agent does not attempt to pick-up the object. Therefore, variability in rewards following a particular action of an agent in a particular state can indicate a potential for cooperation in that state.

Following this intuition, we propose *ensemble value functions for multi-agent exploration* (EMAX), a general framework to seamlessly extend any value-based MARL algorithms by training ensembles of value functions for each agent. To incentivise agents to focus their exploration on state-action pairs that require cooperation across multiple agents, EMAX captures the variability of received rewards via the disagreement of value estimates across the ensemble of value functions. Using an upper-confidence bound (UCB) policy (Auer, 2002), EMAX follows actions with high value estimates and disagreement, corresponding to actions that are considered promising (as measured by the value estimates) and are likely to require cooperation (as measured by the disagreement in value estimates). The EMAX exploration policy can be thought of as an inductive bias for MARL towards exploring state-action pairs that require cooperation. For state-action pairs where little or no cooperation is needed, such as the navigation of the agents in our example, disagreement quickly diminishes. However, for state-action pairs where cooperation is needed, such as picking-up the heavy object, agents will receive variable rewards depending on the actions of other agents. Therefore, the disagreement of value estimates will remain high for such state-action pairs until agents converge towards always succeeding or failing at the task.

Additionally, EMAX leverages the ensemble to compute target values as the average value estimate across the ensemble instead of using target networks. These target values eliminate the need for additional target networks, and have been shown to exhibit lower variance, thereby stabilising the optimisation of agents (Liang et al., 2022). Lastly, actions are selected using a majority vote across the greedy actions of all value functions in the ensemble during evaluation. This evaluation policy reduces the likelihood of suboptimal decision making and, thus, improves the robustness of evaluation performance (Osband et al., 2016a).

We instantiate three value-based MARL algorithms, independent DQN (Mnih et al., 2015), VDN (Sunehag et al., 2018), and QMIX (Rashid et al., 2020), with EMAX and compare them against the corresponding vanilla algorithms in 21 tasks across four diverse multi-agent environments. EMAX improves sample efficiency and final achieved returns across all tasks over all three vanilla algorithms by 60%, 47%, and 538%, respectively (Section 5.2). Additionally, we show that EMAX reduces the variance of gradients throughout optimisation, leading to more stable training, and that the EMAX exploration policy increases the probability of agents to select cooperative actions (Section 5.3). Lastly, we show that comparably small ensembles with five value functions are sufficient to benefit from the advantages of EMAX, discuss the computational cost of ensemble models, and provide ablations to understand the impact of the key ideas of EMAX (Section 5.3).

## 2 Related Work

**Uncertainty for exploration in RL:** Using uncertainty to guide exploration is a well-established idea in RL. One family of algorithms that leverages this idea are randomised value functions (Osband et al., 2019) which are built on the idea of Thompson sampling (Thompson, 1933) from the multi-armed bandits literature (Scott, 2010; Chapelle & Li, 2011). Posterior sampling reinforcement learning extends Thompson sampling by maintaining a distribution of plausible tasks, computes optimal policies for sampled tasks, and continually updates its distribution of tasks from the collected experience (Osband et al., 2013). This approach has extensive theoretical guarantees (Osband & Van Roy, 2017) but does not scale to complex tasks (Osband et al., 2016b). This difficulty has been addressed in subsequent works (Osband et al., 2016b;a; Janz et al., 2019). In particular Bootstrapped DQN (Osband et al., 2016a) approximates randomised value functions by training an ensemble of value functions and explores by randomly sampling a single value function to greedily follow at the beginning of each episode. SUNRISE (Lee et al., 2021) and MeanQ (Liang et al., 2022) also train an ensemble of value functions but instead of sampling value functions to follow for each episode, they follow a UCB policy using the average and standard deviation of value estimates across the ensemble to explore. Moreover, SUNRISE uses the ensemble to weight the value loss based on the variance of target values across the ensemble, and MeanQ stabilises the optimisation by computing lower variance target values as the average value estimate across the ensemble (Anschel et al., 2017). Separately, Fu et al. (2022) extend posterior sampling to model-based RL by learning a probabilistic model of the environment, and Dearden et al. (1998) applied these ideas to tabular Q-learning to learn distributions over Q-values to approximate the value of information for each action. Related to all these ideas, optimistic value estimates in the face of uncertainty can be used to promote exploration for actor-critic (Ciosek et al., 2019) and model-based RL (Sessa et al., 2022). All these approaches make use of uncertainty to guide their exploration, similar to EMAX, but in contrast our work focuses on how to leverage uncertainty to explore and discover cooperation in environments with multiple concurrently learning agents.

**Multi-agent exploration:** For the multi-agent setting, Wang et al. (2020) incentivise agents to interact with each other by intrinsically rewarding them for mutually influencing their transition dynamics or value estimates. Similar intrinsic rewards can be assigned for reaching goal states to train separate exploration policies (Liu et al., 2021). However, intrinsic rewards for exploration have to be carefully balanced for each task due to the modified optimisation objective (Schäfer et al., 2022; Chen et al., 2022). To address this challenge, LIGS (Mguni et al., 2022) formulate the assignment of intrinsic rewards as a MARL problem and train an agent to determine when and which intrinsic reward should be given to each agents. Experience and parameter sharing have been leveraged to greatly improve sample efficiency for MARL by synchronising agents' learning and make use of more data (Christianos et al., 2020; 2021). REMAX (Ryu et al., 2022) identifies valuable initial states for episodes to guide exploration based on a latent representation of states learned using the interactions of agents in the environment. However, there is little research using distributional and ensemble-based techniques for MARL exploration. Zhou et al. (2020) extend posterior sampling (Osband et al., 2013) for MARL, but are limited to two-player zero-sum extensive games. We aim to close this gap by proposing EMAX, an ensemble-based technique for efficient exploration in cooperative MARL. We further highlight that EMAX is a plug-and-play algorithm that can enhance any value-based MARL algorithm, including most existing MARL exploration techniques described in this paragraph.

## 3 Background

### 3.1 Decentralised Partially Observable Markov Decision Process

We formalise cooperative multi-agent environments as decentralised partially observable Markov decision processes (Dec-POMDP) (Pynadath & Tambe, 2002) defined by $(\mathcal{I}, \mathcal{S}, \{\mathcal{A}_i\}_{i \in \mathcal{I}}, \{\mathcal{O}_i\}_{i \in \mathcal{I}}, \mathcal{P}, \mathcal{R}, \Omega)$. Each agent is indexed by $i \in \mathcal{I} = \{1, \ldots, N\}$. $\mathcal{S}$ denotes the state space of the environment. Agents receive local observations which are drawn from their observation space $\mathcal{O}_i$ and take actions from their action space $\mathcal{A}_i$.[1] We denote the space of joint observations and actions across all agents by $\mathcal{O} = \mathcal{O}_1 \times \ldots \times \mathcal{O}_N$ and

---

[1] Since our work focuses on value-based MARL algorithms which extend DQN (Mnih et al., 2015), we assume discrete action spaces for all agents.

$\mathcal{A} = \mathcal{A}_1 \times \ldots \times \mathcal{A}_N$, respectively. The observation function $\Omega : \mathcal{S} \times \mathcal{A} \times \mathcal{O} \mapsto [0,1]$ determines a distribution over joint observations given the current state and taken joint action. Given the current state and the joint action, the transition function $\mathcal{P} : \mathcal{S} \times \mathcal{A} \times \mathcal{S} \mapsto [0,1]$ and reward function $\mathcal{R} : \mathcal{S} \times \mathcal{A} \mapsto \mathbb{R}$ define a distribution over the successor state of the environment and a scalar reward shared across all agents, respectively. Each agent $i$ only receives its local observation $o_i^t \sim \Omega(s^t, a^t)_i$ at timestep $t$ and learns a policy $\pi_i : \mathcal{H}_i \times \mathcal{A}_i \mapsto [0,1]$ defining its action probabilities given the observation history $h_i = (o_i^t)_{t \geq 1} \in \mathcal{H}_i$. The observation history of agent $i$ up to timestep $t$ is defined as $h_i^t = (o_i^\tau)_{\tau=0}^t$. We furthermore denote the joint observation history across all agents by $h = (h_i)_{i \in \mathcal{I}}$, and the joint observation history up to timestep $t$ by $h^t = (h_i^t)_{i \in \mathcal{I}}$. Each agent aims to optimise its policy with the objective of learning a joint policy $\pi = (\pi_1, \ldots, \pi_N)$ such that $\pi \in \arg\max_{\pi'} \mathbb{E}\left[\sum_{t=1}^{\infty} \gamma^{t-1} \mathcal{R}(s^t, a^t)\right]$ with discount factor $\gamma \in [0,1)$.

### 3.2 Value-Based Multi-Agent Reinforcement Learning

**Independent Q-learning:** Independent deep Q-network (IDQN) extends DQN (Mnih et al., 2015) for MARL and independently learns a value function $Q_i$, parameterised by $\theta_i$, for each agent $i$. Agents store tuples $(s^t, h^t, a^t, r^t, s^{t+1}, h^{t+1})$ of experience consisting of state $s^t$, joint observation history $h^t$, applied joint action $a^t$, received reward $r^t$, next state $s^{t+1}$, and next joint observation history $h^{t+1}$, respectively, in a replay buffer. The value function of agent $i$ is then optimised by minimising the average loss across sampled batches of experience:

$$\mathcal{L}(\theta_i) = \left[Q_i(h_i^t, a_i^t; \theta_i) - r^t - \gamma \max_{a_i \in \mathcal{A}_i} \bar{Q}_i(h_i^{t+1}, a_i; \bar{\theta}_i)\right]^2 \tag{1}$$

with $\bar{\theta}_i$ denoting the parameters of the target network $\bar{Q}$ which are periodically copied from $\theta_i$.

**Value decomposition:** Independent learning serves as an effective baseline in many cooperative MARL tasks (Papoudakis et al., 2021) but suffers from several multi-agent challenges such as the multi-agent credit assignment problem, i.e. agents need to identify their individual contribution to received rewards (Du et al., 2019; Rashid et al., 2020), non-stationarity of the optimisation (Papoudakis et al., 2019) and poor sample efficiency in cooperative tasks. Value decomposition algorithms extend IDQN by learning a decomposed centralised state-action value function $Q_{\text{tot}}$, conditioned on the state and joint action of all agents.[2] Directly learning such a value function is often computationally infeasible due to the exponential growth of the joint action space with the number of agents, so the centralised value function is approximated with an aggregation of individual utility functions of all agents conditioned on the local observation history. In this way, value decomposition methods address the multi-agent credit assignment problem since each agent estimates its contribution to the centralised state-action value function with its utility function. The parameterised utility functions and aggregation are optimised by minimising the joint value function loss with target values $y_{\text{tot}}$:

$$\mathcal{L}(\theta) = \left[Q_{\text{tot}}(s^t, a^t; \theta) - y_{\text{tot}}\right]^2 \tag{2}$$

Two common value decomposition algorithms are VDN (Sunehag et al., 2018) and QMIX (Rashid et al., 2020). VDN assumes a linear aggregation of the centralised value function and targets:

$$Q_{\text{tot}}(s^t, a^t; \theta) = \sum_{i \in \mathcal{I}} Q_i(h_i^t, a_i^t; \theta_i) \text{ and } y_{\text{tot}} = r^t + \gamma \max_{a \in \mathcal{A}} \sum_{i \in \mathcal{I}} \bar{Q}_i(h_i^{t+1}, a_i; \bar{\theta}_i) \tag{3}$$

and QMIX assumes a less restrictive monotonic mixing function of individual values:

$$Q_{\text{tot}}(s^t, a^t; \theta) = f_{\text{mix}}\left(Q_1(h_1^t, a_1^t; \theta_1), \ldots, Q_N(h_N^t, a_N^t; \theta_N); \theta_{\text{mix}}\right) \tag{4}$$

$$y_{\text{tot}} = r^t + \gamma \max_{a \in \mathcal{A}} \bar{f}_{\text{mix}}\left(\bar{Q}_1(h_1^{t+1}, a_1; \bar{\theta}_1), \ldots, \bar{Q}_N(h_N^{t+1}, a_N; \bar{\theta}_N); \bar{\theta}_{\text{mix}}\right) \tag{5}$$

with $\theta_{\text{mix}}$ and $\bar{\theta}_{\text{mix}}$ denoting the parameters of the monotonic mixing function $f_{\text{mix}}$ and a delayed target mixing function $\bar{f}_{\text{mix}}$, respectively.

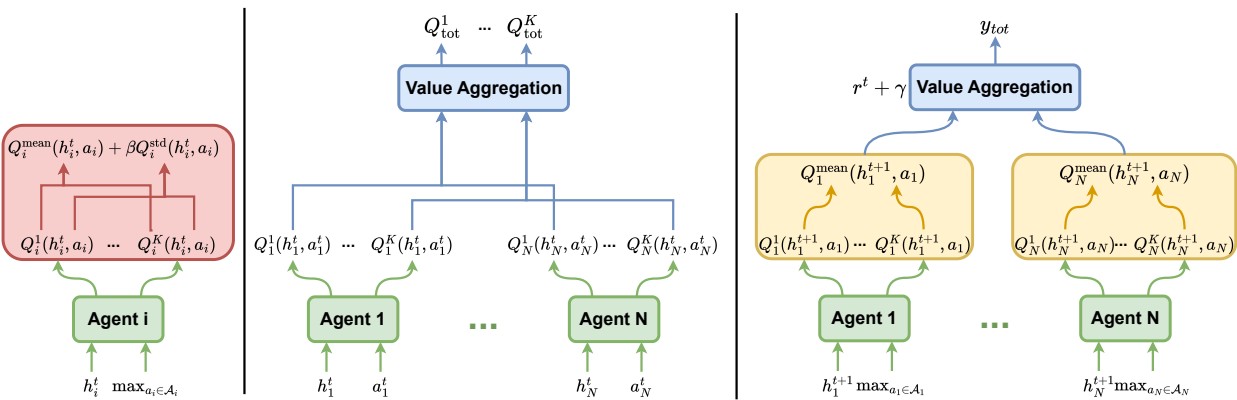

Figure 2: Illustration of EMAX with (left) the UCB exploration strategy for agent $i$, (middle) the computation of value estimates, and (right) the target computation. Computation of individual agent value functions are highlighted in green, exploration in red, value aggregation for value decomposition algorithms in blue, and target aggregation in orange.

## 4 The EMAX Framework: Ensemble Value Functions for Multi-Agent Exploration

In this section, we present ensemble value functions for multi-agent exploration (EMAX), a general framework which trains an ensemble of value functions for each agent in value-based MARL. Formally, each agent $i$ trains an ensemble of $K$ value functions $\{Q_i^k\}_{k=1}^K$ with $Q_i^k$ being parameterised by $\theta_i^k$. Each value function is conditioned on agent $i$'s local observation-action history. EMAX leverages these ensembles of value functions to guide the exploration of agents and stabilise their optimisation, and directly integrates into value decomposition methods such as VDN and QMIX. Figure 2 illustrates the exploration policy and value computation of EMAX, and pseudocode for EMAX is provided in Appendix C.

**Exploration policy:** In multi-agent problems which require agents to cooperate to achieve high returns, agents should focus their exploration on states and actions that require cooperation. To guide agents to explore such states and actions, EMAX agents follow a UCB policy akin to prior work (Lee et al., 2021; Liang et al., 2022) using the average and standard deviation of value estimates across the ensemble:

$$\pi_i^{\text{expl}}(h_i^t; \theta_i) \in \arg\max_{a_i \in \mathcal{A}_i} Q_i^{\text{mean}}(h_i^t, a_i; \theta_i) + \beta Q_i^{\text{std}}(h_i^t, a_i; \theta_i) \tag{6}$$

with the average and standard deviation across the ensemble of value functions of agent $i$ defined as:

$$Q_i^{\text{mean}}(h_i^t, a_i^t; \theta_i) = \frac{1}{K} \sum_{k=1}^K Q_i^k(h_i^t, a_i^t; \theta_i^k) \text{ and } Q_i^{\text{std}}(h_i^t, a_i^t; \theta_i) = \sqrt{\frac{\sum_{k=1}^K \left(Q_i^k(h_i^t, a_i^t; \theta_i^k) - Q_i^{\text{mean}}(h_i^t, a_i^t; \theta_i)\right)^2}{K}} \tag{7}$$

with $\theta_i = \{\theta_i^k\}_{k=1}^K$ denoting the parameters of the ensemble of value functions of agent $i$, and $\beta > 0$ denoting an uncertainty weighting hyperparameter chosen in consideration of the scale of rewards and the amount of exploration required for a task. This policy guides agents to explore actions that are promising (as measured by the mean value estimate) and are likely to require cooperation (as measured by the standard deviation of value estimates). To see why the disagreement of value estimates across the ensemble can indicate whether state-action pairs require cooperation, consider states in which multiple agents have to cooperate, i.e. select specific actions, to receive a large reward. If any agent deviates from the required action, the agents receive no reward. In such states, received rewards for a given action of agent $i$ will vary significantly, since the reward depends on the actions of other agents. In contrast, in states where agent $i$ receives identical rewards independent of the action selection of the other agents, no such variability of rewards is experienced. Due to this variability of rewards (or lack thereof), value estimates across the ensemble will quickly converge in states

---

[2]In environments, where the state is not available during training, we use the joint observation as a proxy for the state.

that require no or limited cooperation, and will exhibit high disagreement in states that require cooperation. Therefore, the EMAX exploration policy focuses the exploration of agents on state-action pairs that require cooperation in contrast to common random exploration for value-based MARL such as $\epsilon$-greedy policies. We empirically demonstrate these benefits in Section 5.3.

**Optimisation:** To extend IDQN with EMAX, we optimise the $k$-th value function of agent $i$ by minimising the following loss:

$$\mathcal{L}(\theta_i^k) = \left[ Q_i^k(h_i^t, a_i^t; \theta_i^k) - r^t - \gamma \max_{a_i \in \mathcal{A}_i} Q_i^{\mathrm{mean}}(h_i^{t+1}, a_i; \theta_i) \right]^2 \tag{8}$$

Computing target values as the average across all value estimates of the ensemble (Liang et al., 2022) reduces the computational and memory cost of training ensemble networks by eliminating the need for target networks and, as we empirically show in Section 5.3, reduces the variability of gradients. Such reduced variability of gradients improves the stability of training. Furthermore, we argue that such improved stability of gradients during the optimisation is particularly valuable in MARL where the exploration and non-stationarity of the policies of other agents can otherwise result in unstable training.

**Value decomposition:** As defined above, the EMAX exploration policy guides agents towards states and actions in which cooperation is required, but it does not distinguish which agents need to cooperate in a particular state. For example, some of the agents might need to cooperate in a state to receive a large reward but other agents do not contribute to this reward. To overcome this limitation such that the EMAX exploration policy guides each agent towards states and actions in which that agent's cooperation rather than any agents' cooperation is required, it is important that the value estimates of that agent correspond to its individual contribution to the common rewards.

Value decomposition techniques such as VDN (Sunehag et al., 2018) and QMIX (Rashid et al., 2020) are designed to learn individual value functions for each agent that identify their contribution to received common rewards. By integrating these techniques into EMAX, the exploration policy can benefit from the multi-agent credit assignment achieved by the value decomposition, and the optimisation of the value functions can be stabilised by computing target estimates across the ensemble as proposed in EMAX. To integrate value decomposition methods into EMAX, each agent trains an ensemble of independent value functions as proposed above. The total loss for the $k$-th value functions of all agents with parameters $\theta^k = \{\theta_i^k\}_{i \in \mathcal{I}}$ is given by Equation (2) with centralised value function and targets for VDN (Equation (9)):

$$Q_{\mathrm{tot}}^k(s^t, a^t; \theta^k) = \sum_{i \in \mathcal{I}} Q_i^k(h_i^t, a_i^t; \theta_i^k) \text{ and } y_{\mathrm{tot}} = r^t + \gamma \max_{a \in \mathcal{A}} \sum_{i \in \mathcal{I}} Q_i^{\mathrm{mean}}(h_i^{t+1}, a_i; \theta_i) \tag{9}$$

and QMIX (Equations (10) and (11)):

$$Q_{\mathrm{tot}}^k(s^t, a^t; \theta^k) = f_{\mathrm{mix}}\left( Q_1^k(h_1^t, a_1^t; \theta_1^k), \ldots, Q_N^k(h_N^t, a_N^t; \theta_N^k); \theta_{\mathrm{mix}} \right) \tag{10}$$

$$y_{\mathrm{tot}} = r^t + \gamma \max_{a \in \mathcal{A}} \bar{f}_{\mathrm{mix}}\left( Q_1^{\mathrm{mean}}(h_1^{t+1}, a_1; \theta_1), \ldots, Q_N^{\mathrm{mean}}(h_N^{t+1}, a_N; \theta_N); \bar{\theta}_{\mathrm{mix}} \right) \tag{11}$$

For QMIX, we use a single mixing network and target mixing network with parameters $\theta_{\mathrm{mix}}$ and $\bar{\theta}_{\mathrm{mix}}$, respectively, to aggregate the value estimates for all value functions in the ensemble. The aggregation of QMIX is able to represent a wider set of centralised value functions, but VDN has been shown to be more sample efficient in tasks which do not seem to require a non-linear aggregation (Papoudakis et al., 2021). Therefore, we consider both the extension of VDN and QMIX with EMAX.

**Evaluation policy:** When evaluating agents, value-based MARL algorithms typically follow the greedy policy with respect to their value function. With EMAX, agent $i$ selects its action during evaluation using a majority vote across the greedy actions of all models in its ensemble:

$$\pi_i^{\mathrm{eval}}(h_i^t; \theta_i) \in \arg\max_{a_i \in \mathcal{A}_i} \sum_{k=1}^K [1]_{\mathcal{A}_{\mathrm{opt},i}^k}(a_i) \text{ and } \mathcal{A}_{\mathrm{opt},i}^k = \{ a_i \in \mathcal{A}_i \mid a_i \in \arg\max_{a_i'} Q_i^k(h_i^t, a_i'; \theta_i^k) \} \tag{12}$$

with indicator function $[1]_{\mathcal{A}_{\mathrm{opt},i}^k}(a) = 1$ for the greedy action(s) of the $k$-th value function of agent $i$, $a_i \in \mathcal{A}_{\mathrm{opt},i}^k$, and 0 otherwise. Such a policy decreases the likelihood of taking poor actions because any individual value

function preferring a poor action due to errors in value estimates does not impact the action selection as long as the majority of models agree on the optimal action.

**Ensemble value functions** Motivated by previous work in cooperative MARL that shares networks across agents to improve sample efficiency and scalability (e.g. (Papoudakis et al., 2021; Christianos et al., 2021; Albrecht et al., 2024)), and the computational cost of training $K$ value functions for each agent, we share a single ensemble of value functions across all agents. All aforementioned techniques rely on value functions within the ensemble to be sufficiently diverse, in particular early in training. To ensure such diversity, we implement the $K$ value functions within the ensemble as entirely separate networks with no sharing of parameters across the value functions in the ensemble. To efficiently compute the value estimates of all value functions and all agents in a single forward pass, we vectorise the computation across agents and networks within the ensemble. Beyond separate networks within the ensemble, we employ three techniques from prior work (Osband et al., 2016a; Liang et al., 2022) to incentivise diversity of value functions within the ensemble: (1) Ensemble models are separately and randomly initialised. (2) We sample separate batches of experience from the replay buffer to train each model in the ensemble.(3) Each model is trained on bootstrapped samples of the entire experience collected. For more details on the bootstrapped sampling process, see Appendix C.1.

## 5 Experiments

### 5.1 Evaluation Details

We evaluate a total of eleven deep MARL algorithms: Independent DQN (IDQN), VDN, and QMIX as well as their extensions with EMAX, which we will denote IDQN-EMAX, VDN-EMAX, and QMIX-EMAX, respectively, three value-based exploration algorithms in MAVEN (Mahajan et al., 2019), CDS (Li et al., 2021), and EMC (Zheng et al., 2021), and independent and multi-agent PPO (IPPO and MAPPO) which have been shown to exhibit strong MARL performance (Papoudakis et al., 2021; Yu et al., 2022). Following Agarwal et al. (2021), we report performance profiles and use the interquartile mean (IQM) and 95% confidence intervals computed over five runs in all tasks. Learning curves indicate the sample efficiency of agents which is largely determined by their ability to effectively explore the environment, and performance profiles allow to compare algorithms with respect to the robustness of their final policies. For every algorithm and task, agents share network parameters with networks receiving the agent identity in the form of a onehot vector as additional input. Unless stated otherwise EMAX uses ensembles with $K = 5$ value functions. Details on computational resorces and hyperparameters are provided in Appendices A and D, respectively. We evaluate in 21 tasks across four multi-agent environments, visualised in Figure 3: eight level-based foraging (LBF) tasks (Albrecht & Ramamoorthy, 2013; Papoudakis et al., 2021), four boulder-push (BPUSH) tasks (Christianos et al., 2022), six multi-robot warehouse (RWARE) tasks (Christianos et al., 2020; Papoudakis et al., 2021), and three multi-agent particle environment (MPE) tasks (Mordatch & Abbeel, 2018; Lowe et al., 2017). These tasks were selected since they represent a diverse set of cooperative MARL tasks which require agents to cooperate to achieve high rewards. Many of these tasks are further considered challenging since rewards are sparse such that effective exploration is essential to enable efficient learning. We briefly describe each environment below. More details on each environment can be found in Appendix B.

**Level-based foraging:** The level-based foraging (LBF) environment (Albrecht & Ramamoorthy, 2013; Papoudakis et al., 2021) contains diverse tasks in which agents and food are randomly scattered in a gridworld. Agents and food items are assigned levels, and agents can only pick-up food if the collective level of all agents that are standing next to the food and chose the pick-up action is greater or equal to the level of the food. Therefore, agents need to cooperate to pick-up food with high level, while agents can pick-up food items with low levels individually. Tasks vary in the size of the gridworld, the number of agents and food, and the level assignment.

**Boulder-push:** In the boulder-push environment (BPUSH) (Christianos et al., 2022), agents need to navigate a gridworld to move a boulder to a target location. To successfully move the boulder towards its target location, all agents need to stand next to the boulder and move against it in the same direction at the same timestep. Agents are rewarded for pushing the boulder forward but receive a small negative reward for any miscoordination, i.e. any but not all agents move against the boulder. We consider BPUSH tasks with varying sizes of the gridworld and the number of agents varying between two and four.

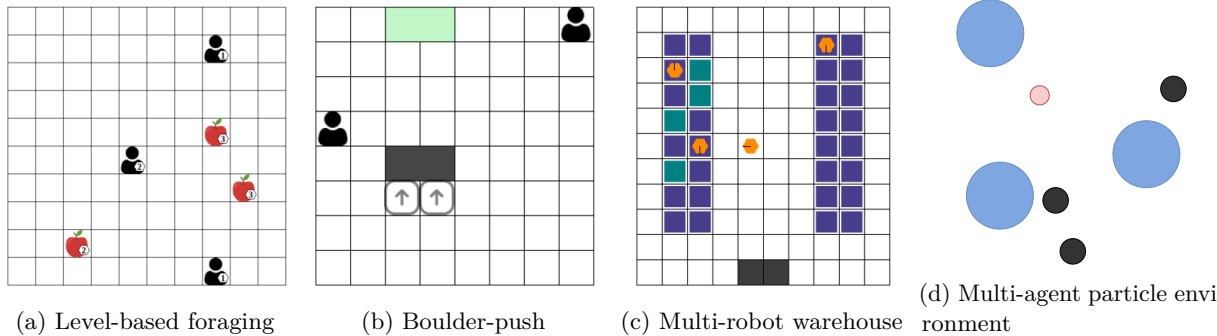

(a) Level-based foraging      (b) Boulder-push      (c) Multi-robot warehouse    (d) Multi-agent particle environment

Figure 3: Visualisations of four multi-agent environments.

**Multi-robot warehouse:** The multi-robot warehouse environment (RWARE) (Christianos et al., 2020; Papoudakis et al., 2021) represents gridworld warehouses with blocks of shelves. Agents need to navigate the warehouse and collect currently requested items. Agents only observe nearby agents and shelves immediately next to their location, and are only rewarded for successfully delivering requested shelves, which requires a long sequence of specific actions. This results in very sparse rewards which makes RWARE tasks hard exploration problems. It is worth highlighting that no value-based algorithm achieved non-zero rewards in this environment within four million timesteps of training in prior evaluations (Papoudakis et al., 2021).

**Multi-agent particle environment:** In the multi-agent particle environment (MPE) (Mordatch & Abbeel, 2018; Lowe et al., 2017), agents navigate continuous two-dimensional, fully-observable environments. We evaluate agents in three diverse tasks which all require cooperation between agents with dense rewards. (1) Predator-prey in which three agents control predators in an environment with three landmarks, representing obstacles, and a faster, pre-trained prey. (2) Spread in which three agents need to cover three landmarks while avoiding collisions with each other. (3) Adversary in which two agents are in an environment with an pre-trained adversary and two landmarks. At the beginning of each episode, one of the two landmarks is randomly determined as the goal landmark for the agents (agents observe this goal landmark but the adversary has no information about it). Both agents are rewarded for one of them being close to the goal landmark but they are negatively rewarded for the adversary agent moving close to the goal location.

## 5.2 Evaluation Results

Figure 4 visualises the learning curve and performance profile of evaluation returns of all algorithms across all 21 tasks. Across all tasks, EMAX improves final evaluation returns of IDQN, VDN, and QMIX, shown in Figure 4a, by 60%, 47%, and 538%, leading to higher final returns compared to their vanilla baselines in 18, 16, and 20 out of 21 tasks, respectively. These results arise from EMAX improving the sample efficiency and learning stability of the vanilla algorithms, as we will show in Section 5.3. Additionally, QMIX-EMAX is able to learn effective policies in several hard exploration tasks where QMIX fails to achieve any reward. The performance profile in Figure 4b visualises the the distribution of evaluation returns at the end of training across all algorithms and tasks. These profiles indicate that EMAX significantly improves the robustness of all algorithms, consistently achieving higher returns. We provide normalised evaluation returns for each environment, as well as learning curves in all individual tasks in Appendices E and F, respectively.

In LBF, EMAX significantly improves the performance of QMIX whereas minor improvements can be seen for IDQN and VDN. Inspecting learning curves in individual tasks (Appendix F.1) shows that QMIX, MAVEN, CDS and EMC fail to achieve any rewards in several LBF tasks with particularly sparse rewards. We hypothesise that these algorithms suffer from the large dimensionality of the joint observation as input to the mixing network which is inefficient to train with the sparse learning signal of these tasks. The uncertainty-guided exploration of EMAX seems to alleviate these inefficiencies.

In BPUSH, a similar trend can be observed. Most notably, VDN-EMAX and QMIX-EMAX learn to solve a BPUSH task with four agents in which no baseline demonstrates any positive rewards (see Figure 11d). This

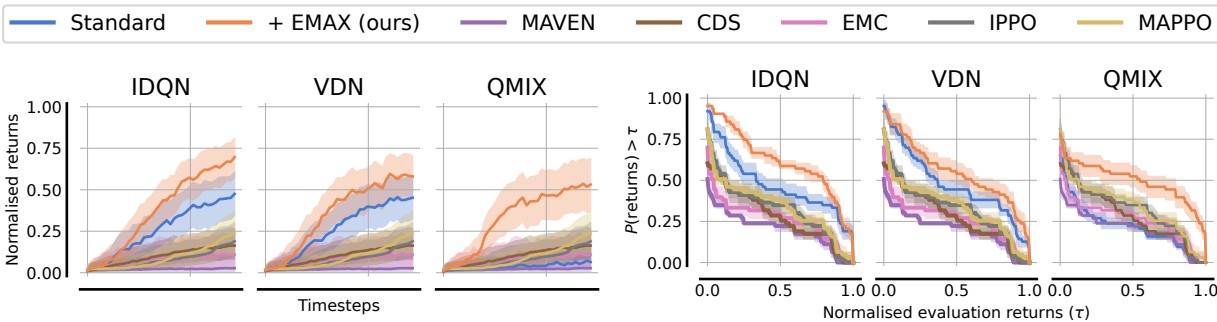

(a) Learning curve of normalised evaluation returns    (b) Performance profile of final evaluation returns

Figure 4: (a) Evaluation returns throughout training and (b) performance profile (Agarwal et al., 2021) visualising the distribution of evaluation returns at the end of training of all algorithms, both aggregated across all 21 tasks. EMAX (orange) significantly improves the sample efficiency and final achieved returns of all algorithms. Lines and shading represent the interquartile mean and 95% confidence intervals of evaluation returns, respectively, aggregated over five runs for every task, for a total of 105 runs per algorithm. For each task, evaluation returns are normalised between the minimum (0) and maximum (1) achieved returns.

task requires complex cooperation because four agents need to move in unison to successfully complete this task and any miscoordination leads to negative rewards.

In RWARE, consistent with prior work (Papoudakis et al., 2021), independent learning value-based algorithms outperform centralised value decomposition methods due to highly sparse rewards. IDQN-EMAX outperforms all baselines across four RWARE tasks with larger warehouses, and IDQN-EMAX and VDN-EMAX both significantly improve upon their vanilla baselines in all RWARE tasks, achieving 330% and 252% higher final evaluation returns, respectively, whereas QMIX with and without EMAX fail to learn. Lastly, IPPO and MAPPO perform well in RWARE with MAPPO reaching highest evaluation returns in two smaller RWARE tasks. However, in tasks with larger warehouses and, thus, more required exploration, IDQN-EMAX outperforms all other algorithms. To the best of our knowledge, this is the first time that any value-based algorithm outperforms actor-critic methods like IPPO and MAPPO in RWARE tasks.

In contrast to other environments, MPE has continuous observations and dense rewards. In all three MPE tasks, we see improvements in sample efficiency and final performance for algorithms with EMAX compared to vanilla algorithms. In particular in the MPE spread task, EMAX significantly improves the performance of all extended algorithms by significant margins.

## 5.3 Analysis and Ablations

In this section, we further investigate the efficacy of all components of EMAX to study our hypotheses from Section 4 and answer the following questions: (1) Do EMAX target values improve the stability of training? (2) Does the EMAX exploration policy lead to more exploration of states and actions with potential for cooperation? (3) Does the EMAX evaluation policy reduce the likelihood of selecting sub-optimal actions? After answering these questions, we provide further ablations of these techniques and show that comparably small ensembles of value functions are sufficient to achieve the benefits of EMAX.

**Training stability:** In Section 4, we stated that the computation of EMAX target values reduces the variability of gradients during training, and, thus, improves stability of the optimisation (as previously observed in single-agent RL (Liang et al., 2022)). To demonstrate this stabilising effect, we visualise the stability of gradients measured by the conditional value at risk (CVaR) of gradient norms, detrended over consecutive values, during the optimisation of IDQN, VDN, QMIX with and without EMAX

$$\text{CVaR}(g') = \mathbb{E}\left[g' \mid g' \geq \text{VaR}_{95\%}(g')\right] \text{ and } g'_t = |\nabla_{t+1}| - |\nabla_t| \tag{13}$$

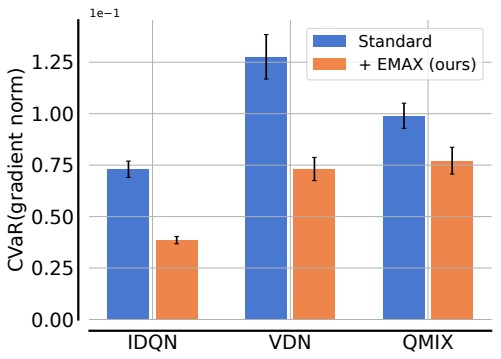

(a) Average and standard error of CVaR values across all tasks with and without EMAX.

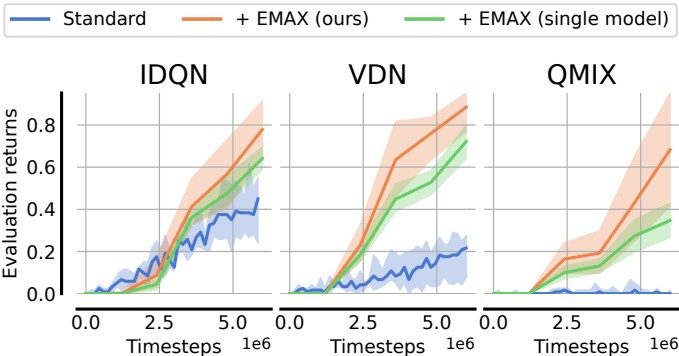

(b) Evaluation policy ablation in LBF 10x10-4p-4f-coop.

Figure 5: (a) Average and standard error of the conditional value at risk (CVaR) of detrended consecutive gradient norms of all algorithms across all tasks. This metric corresponds to the short-term risk across time suggested by Chan et al. (2020). (b) Evaluation returns for the vanilla and EMAX algorithms in the LBF 10x10-4p-4f-coop task, and an ablation of the evaluation policy. For the single model ablation, the agents follow the greedy policy with respect to a single value function within their model instead of computing a majority vote across greedy policies.

where the value at risk (VaR) corresponds to the value at the 95% quantile of all detrended gradient norm values. This metric indicates the stability of gradients throughout training. For example, a larger CVaR value indicates more variability in gradients which can indicate unstable training, while a smaller CVaR value indicates less variability in gradients and more stable training. Figure 5a shows the average and standard error of these CVaR values across all 21 tasks. We observe that the target computation of EMAX significantly reduces the CVaR of gradient norms for IDQN, VDN, and QMIX indicating more stable optimisation, thus, confirming our hypothesis. The difference for QMIX is less pronounced since the base algorithm fails to learn in several tasks, leading to little training with low gradient variability independent of the target values.

**Exploration policy:** To validate our hypothesis that the EMAX exploration policy leads to more exploration of states and actions with the potential for cooperation (Section 4), we train IDQN, VDN, and QMIX with and without EMAX in the LBF 10x10-3p-5f task with a large quantity of food, some of which require cooperation and some of which can picked-up by individual agents. Figure 6 shows the evaluation returns throughout training, the average distances of agents to the closest food, and the percentage of agents selecting the pick-up action in states where multiple agents need to cooperate to pick-up food.[3] We emphasise that a lower average distance of agents to the closest food and a higher percentage of selecting the pick-up action in cooperative states indicates that agents seek out states with the potential for cooperation and prefer to apply actions with the chance for cooperation, respectively. Figure 6 shows that our hypotheses about the EMAX exploration policy hold in the tested task. (1) EMAX leads to agents effectively seeking out food as states with chance for cooperation, as indicated by the lower average distance of EMAX compared to the baseline in Figure 6b, and (2) EMAX agents select the pick-up action in such cooperative states more often, as shown in Figure 6c, resulting in significantly higher evaluation returns. To separate the effect of the efficacy of the EMAX training and the exploration policy, we also compare to the percentage of choosing to pick-up in cooperative states by greedily following any of the value functions in the ensemble instead of following the UCB policy across the ensemble. While this ablation leads to a significant improvement over the vanilla algorithms, it still exhibits a lower rate of choosing to cooperate compared to the EMAX exploration policy. Lastly, Figure 6d visualises the mean and standard deviation of action-value estimates across the ensemble for the no-op action, movement actions, and the pick-up action in states with the potential for cooperation between agents. As we can see, the mean and standard deviation of the action-values for the pick-up action

---

[3]To obtain the average distances to nearby food and cooperation percentages, we rollout the exploration policy of the baseline algorithm and the EMAX UCB exploration policy in the LBF task for 50 episodes every 200,000 timesteps of training, and compute the respective values across rollouts.

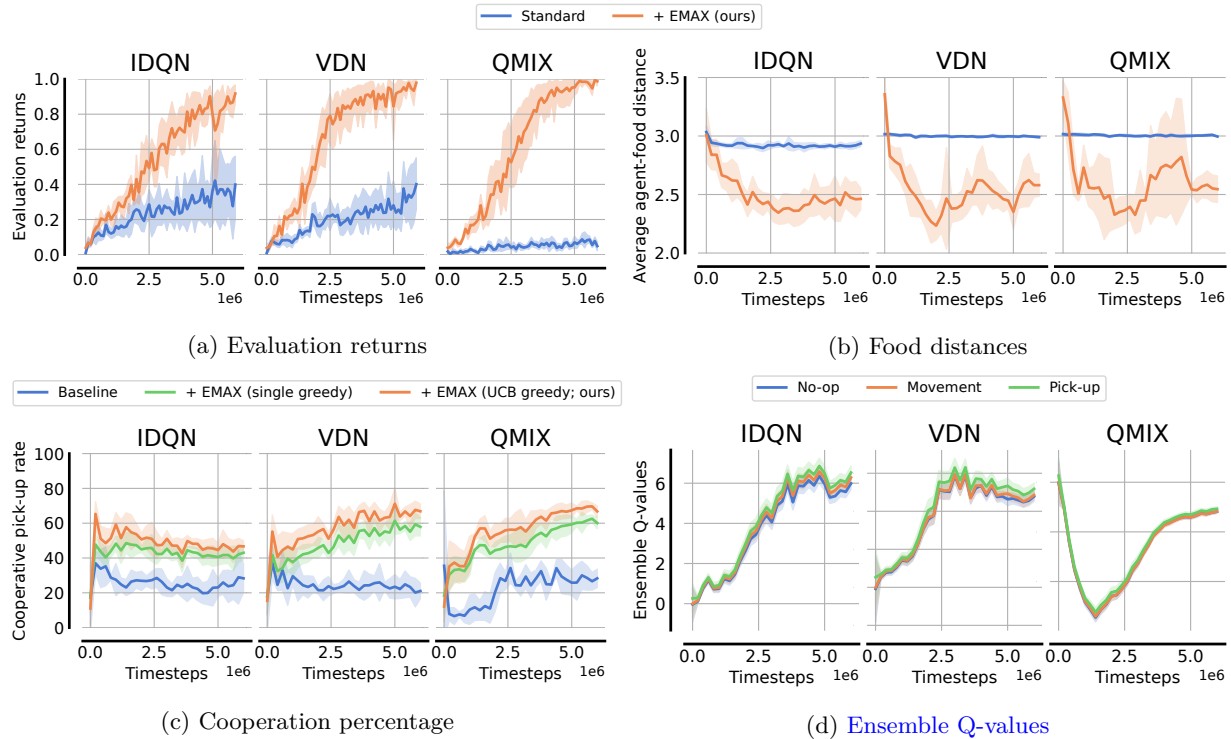

(a) Evaluation returns

(b) Food distances

(c) Cooperation percentage

(d) Ensemble Q-values

Figure 6: (a) Mean and 95% bootstrapped confidence intervals of evaluation returns, (b) mean and standard deviation of average food distances across rollouts, (c) mean and standard deviation of percentages of agents selecting the pick-up action and in states with a chance for cooperation, and (d) mean and standard deviation of action-value estimates for no-op, movement, and pick-up actions in states with a chance for cooperation in the LBF 10x10-3p-5f task.

are larger compared to other actions, indicating that the EMAX value estimates correlate with the potential for cooperation of this action.

**Evaluation policy:** The evaluation policy of EMAX selects actions by a majority vote across all policies in the ensemble (Equation (12)). We hypothesised that such a policy is more robust to sub-optimal action selection since any individual policy taking sub-optimal actions does not impact the executed policy as long as the majority of policies agree on the optimal action. Figure 5b shows the evaluation returns of IDQN, VDN, and QMIX with and without EMAX in the LBF 10x10-4p-4f-coop task. For EMAX, we show the evaluation policy using majority voting (ours) as well as an ablation following the greedy policy with respect to any of the individual value functions within the ensemble (single model). We highlight that no further agents were trained, but we directly extract the individual value functions within the ensemble and evaluate them, so the only difference in the EMAX single policy ablation and ours is the followed policy. This experiment indicates the improved robustness of our majority voting to select actions leading to higher evaluation returns in a task which frequently requires agents to cooperate.

**Ensemble size:** The computational cost of training an ensemble of models scales with the ensemble size $K$. To illustrate the additional cost, we investigate the training speed of EMAX for varying $K$ and pose the question of how many models are needed in the ensemble to benefit from the improved training stability and exploration. To answer this question, we evaluate all algorithms with varying $K$ in the RWARE 11x10 task with four agents (Figure 7), in which EMAX led to substantial improvements for IDQN and VDN. It appears that the benefits of larger ensemble models saturate at $K = 5$. EMAX with $K = 8$ performs identical or worse for all algorithms, and the smaller ensemble $K = 2$ reaches lower returns for IDQN and VDN. These results suggest that a comparably small ensemble with $K = 5$ can significantly improve sample efficiency with EMAX. Additionally, we hypothesise that larger ensemble value functions may require more data to

| Algorithm | Baseline | $K = 2$ | $K = 5$ | $K = 8$ |
|-----------|----------|---------|---------|---------|
| IDQN | 16.80 | 21.29 (+27%) | 33.04 (+97%) | 48.06 (+186%) |
| VDN | 16.92 | 21.56 (+27%) | 33.25 (+97%) | 48.16 (+185%) |
| QMIX | 17.70 | 22.53 (+27%) | 33.71 (+90%) | 48.66 (+175%) |

Table 1: Average training time (in seconds) for vanilla and EMAX algorithms with varying ensemble sizes $K$ to complete 10,000 timesteps of training in the LBF 10x10-3p-3f task. Relative increase to the training time of the baseline algorithm ($K = 1$) is given in parenthesis. Times are averaged across ten runs.

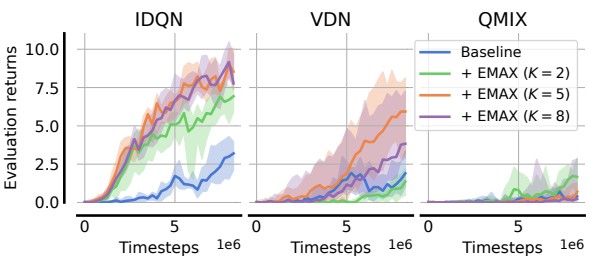

Figure 7: Evaluation returns for varying ensemble sizes $K \in \{2, 5, 8\}$ in the RWARE $11 \times 10$ 4ag task.

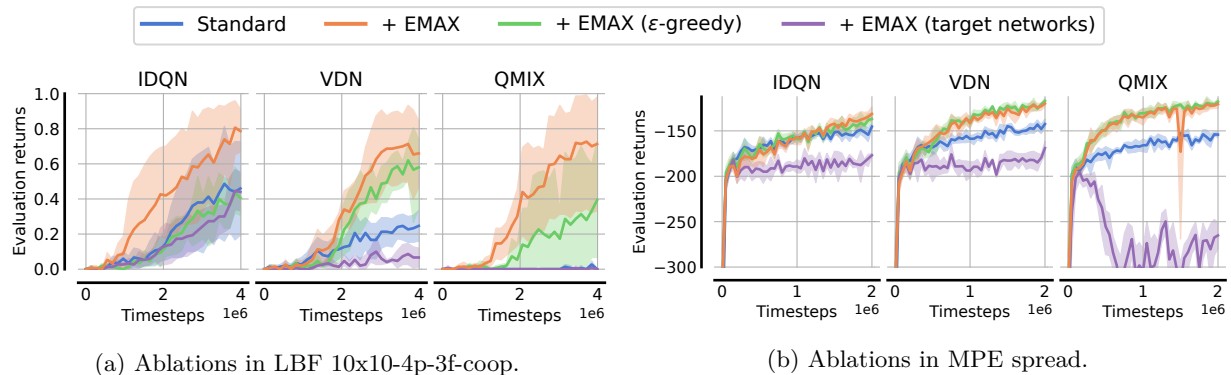

(a) Ablations in LBF 10x10-4p-3f-coop.

(b) Ablations in MPE spread.

Figure 8: Mean and 95% confidence intervals of evaluation returns for all vanilla and EMAX algorithms and two ablations in (a) LBF 10x10-4p-3f-coop, and (b) MPE spread. For the ablations, we replace the UCB exploration policy with $\epsilon$-greedy exploration (green) and the EMAX target computation with standard target networks (purple), respectively.

train, thus leading to diminishing benefits for ensembles of many value functions. Table 1 shows the average time to train IDQN, VDN, QMIX, and their corresponding EMAX extensions with $K \in \{2, 5, 8\}$ for 10,000 timesteps in the LBF 10x10-3p-3f task. These times were averaged across ten runs. We can see that training an ensemble of $K = 5$ value functions, as applied in our evaluation, increases the training time by less than 100%. While this cost is significant, the increase in computational cost is notably less than a linear increase due to parallelisation on modern hardware, and we believe that it is justified in cases where the significant improvements of EMAX in both sample efficiency and stability are of importance. In Appendix G, we also show that EMAX still outperforms the vanilla algorithms when roughly matching or even exceeding the number of parameters of the ensemble models of EMAX.

**Ablations:** We already demonstrated that the EMAX target computation and exploration policy lead to more robust gradients during optimisation and focus exploration on cooperative actions. To discriminate the importance of both of these components to the performance of EMAX algorithms, we conduct an ablation study in two tasks: LBF 10x10-4p-3f-coop, and MPE spread. We compare the performance of the full EMAX algorithm to two ablations which (1) substitute the UCB exploration policy with an $\epsilon$-greedy exploration policy, and (2) use target networks to compute target values with each network in the ensemble having its own target network. The results of this ablation study (Figure 8) demonstrate the importance of both of these components, leading to notably better or similar performance for all algorithms in both tasks. In particular the EMAX target computation significantly improves performance across all algorithms and tasks. The UCB exploration policy leads to significant improvements in LBF, but only marginal gains in MPE. We hypothesise that agents need to explore less in the MPE spread task due to the dense rewards, and that there are only few states in which agents have to apply a particular action to coordinate.

# 6 Conclusion

In this paper, we proposed EMAX, a general framework to extend any value-based MARL algorithms using ensembles of value functions. EMAX leverages the disagreement of value estimates across the ensemble with a UCB policy to guide exploration towards parts of the environment which require coordination. Additionally, gradients during training are stabilised by computing target values as the average value estimate across the ensemble. Empirical results in 21 tasks across four environments demonstrated that EMAX significantly improves sample efficiency, final performance, and training stability for all three extended algorithms. Lastly, we discussed the computational cost introduced by EMAX and showed that comparably small ensemble models are sufficient to achieve the demonstrated improvements.

EMAX is currently limited to value-based cooperative MARL algorithms. Firstly, future work should consider the extension of EMAX to multi-agent actor-critic algorithms such as MAPPO and IPPO, which have shown to be effective in cooperative MARL (Yu et al., 2022). Ensembles of critics and policies could be trained for each agent, with similar target computation and UCB policies across actors being used to leverage the techniques proposed in this work. Secondly, future work could aim to reduce the computational cost of training ensembles of value functions. Prior work has explored the application of hypernetworks (Dwaracherla et al., 2020) and latent-conditioned models (Shen & How, 2023) to approximate ensembles using a single network. Similar techniques could help to significantly reduce the computational cost of EMAX, thereby making it more widely accessible. Lastly, ensembles of value functions can be used to efficiently explore in two-player zero-sum games (Perolat et al., 2022; McAleer et al., 2023; Sokota et al., 2022).

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

## A    Computational Resources

All gridsearches and evaluations for deep experiments were conducted on (1) desktop computers with two Nvidia RTX 2080 Ti GPUs, Intel i9-9900X @ 3.50GHz CPU, 62GB RAM, running Ubuntu 20.04, and (2) two server machines with four Nvidia V100 GPUs, Intel Xeon Platinum 8160 @ 2.10GHz CPU, 503GB RAM, running CentOS Linux 7 OS. The speedtest for varying ensemble sizes reported in Table 1 has been conducted on the desktop computer. For experiments, we use Python 3.9.13 with PyTorch 1.12.1, and NumPy 1.23.3.

## B    Environments

**Level-Based Foraging:** The level-based foraging (LBF) environment (Albrecht & Ramamoorthy, 2013; Papoudakis et al., 2021) contains diverse tasks in which agents and food are randomly scattered in a gridworld. Agents observe the location of themselves as well as all other agents and food in the gridworld, and are able to choose between discrete actions $\mathcal{A} = \{\text{do nothing, move up, move down, move left, move right, pick-up}\}$. Agents and food are assigned levels and agents can only pick-up food if the level of all agents standing next to the food and choosing the pick-up action together is greater or equal to the level of the food. Agents only receive rewards for successful collection of food. Episodes terminate after all food has been collected or after at most 50 timesteps. Each episode randomises the level and starting locations of agents and food. Tasks vary in the size of the gridworld, the number of agents and food, and the level assignment.

**Boulder-Push:** In the boulder-push environment (BPUSH) (Christianos et al., 2022), agents need to navigate a gridworld to move a boulder to a target location. Agents observe the location of the boulder, all other agents, and the direction the boulder needs to be pushed in. The action space of all agents consists of the same discrete actions $\mathcal{A} = \{\text{move up, move down, move left, move right}\}$. Agents only receive rewards of 0.1 per agent for successfully pushing the boulder forward in its target direction, which requires cooperation of all agents, and a reward of 1 per agent for the boulder reaching its target location. Unsuccessful pushing of the boulder by some but not all agents leads to a penalty reward of $-0.01$. Episodes terminate after the boulder reached its target location or after at most 50 timesteps. BPUSH tasks considered in this work vary in the size of the gridworld and the number of agents varying between two and four.

**Multi-Robot Warehouse:** The multi-robot warehouse environment (RWARE) (Christianos et al., 2020; Papoudakis et al., 2021) represents gridworld warehouses with blocks of shelves. Agents need to navigate the warehouse and collect currently requested items. Agents only observe nearby agents and shelves immediately next to their location, and choose discrete actions $\mathcal{A} = \{\text{turn left, turn right, move forward, load/ unload shelf}\}$. Agents are only rewarded for successful deliveries of requested shelves, which require long sequences of actions, with a reward of 1, thus rewards are very sparse making RWARE tasks hard exploration problems. At each timestep, the total number of requested shelves is equal to the number of agents and once requested shelves, a currently unrequested shelf is uniformly at random sampled and added to the list of requested shelves. Episodes terminate after 500 timesteps. While agents are capable of delivering shelves without interaction with other agents, agents need to cooperate to avoid blocking each others path in narrow parts of the warehouse and trying to move to and load identical requested shelves. To maximise episodic returns, agents need to learn to avoid such conflicts with other agents which requires them to learn conventions and cooperate. It is worth highlighting that no value-based algorithm achieved non-zero rewards in this environment within four million timesteps of training in prior evaluations (Papoudakis et al., 2021).

**Multi-Agent Particle Environment:** In the multi-agent particle environment (MPE) (Mordatch & Abbeel, 2018; Lowe et al., 2017), agents navigate continuous two-dimensional, fully-observable environments. In all tasks, agents observe the relative position and velocity of all agents, as well as the relative positions of landmarks in the environment. Agents choose between five discrete actions consisting of doing nothing and movement in all four cardinal directions. We evaluate agents in three diverse tasks within MPE which all require cooperation between all agents with densely rewarded objectives. (1) Predator-prey in which three agents control predators in an environment with three landmarks, representing obstacles, and a faster, pre-trained[4] prey. (2) Spread in which three agents need to cover three landmarks while avoiding collisions

---

[4]Pre-trained agents are obtained from the EPyMARL codebase (Papoudakis et al., 2021). They were obtained by training all agents (including adversaries) with the MADDPG algorithm for 25,000 episodes.

with each other. (3) Adversary in which two agents are in an environment with an pre-trained adversary and two landmarks. At the beginning of each episode, one of the two landmarks is randomly determined as the goal landmark for the agents (agents observe this goal landmark but the adversary has no information about it). Both agents are rewarded for one of them being close to the goal landmark but they are negatively rewarded for the adversary agent moving close to the goal location. Therefore, agents are incentivised to cover both landmarks in order to maximise their rewards since then they are receiving rewards for covering the goal landmark but also preventing the adversary agent from identifying which landmark is the goal landmark.

## C   Algorithmic Details

In this section, we provide pseudocode for training IDQN with the EMAX extension (Algorithm 1), for training QMIX with the EMAX extension (Algorithm 3), and for evaluating any algorithm with EMAX (Algorithm 2). The pseudocode for VDN with EMAX is analogous to the pseudocode for QMIX with EMAX without the mixing network and computing total value estimates and targets in the loss computation by following Equation (9). Furthermore, we provide additional details on the bootstrapped sampling process in Appendix C.1.

---

**Algorithm 1** Training IDQN with EMAX

Initialise parameters $\{\theta_i^k\}_{k=1}^K$ of value functions $\{Q_i^k\}_{k=1}^K$ for each agent $i \in \mathcal{I}$
Initialise empty episodic replay buffer $\mathcal{D}$
**for** each episode **do**
    Obtain initial state $s^0$ and joint observation $o^0$
    **for** each step $t = 0, \cdots, T$ **do**
        **for** each agent $i \in \mathcal{I}$ **do**
            Select action $a_i^t \sim \pi_i^{\text{expl}}(h_i^t; \theta_i)$ (Equation (6))
        **end for**
        Apply joint action $a^t = (a_1^t, \ldots, a_{|\mathcal{I}|}^t)$
        Receive next state $s^{t+1} \sim \mathcal{P}(s^t, a^t)$, reward $r^t = \mathcal{R}(s^t, a^t)$, and joint observation $o^{t+1} \sim \Omega(s^t, a^t)$
    **end for**
    Sample bootstrap masks $\{m^k\}_{k=1}^K$ from Bernoulli($p$)
    Store episode $(s^{0:T}, h^{0:T}, a^{0:T}, r^{0:T}, \{m_k\}_{k=1}^K)$ in $\mathcal{D}$
    **for** each agent $i \in \mathcal{I}$ **do**
        **for** each model in the ensemble $k = 1, \ldots, K$ **do**
            Sample batch of episodes $\mathcal{B}$ from $\mathcal{D}$ with $m_k = 1$
            Compute the average loss $\mathcal{L}(\theta_i^k)$ (Equation (8)) for each timestep $t$ in $\mathcal{B}$
            Update $\theta_i^k$ by minimising $\mathcal{L}(\theta_i^k)$
        **end for**
    **end for**
**end for**

---

**Algorithm 2** Evaluating with EMAX

**Require:** Trained ensemble of value functions $\{Q_i^k\}_{k=1}^K$ for each agent $i \in \mathcal{I}$
    Obtain initial state $s^0$ and joint observation $o^0$
    **for** each step $t = 0, \cdots, T$ **do**
        **for** each agent $i \in \mathcal{I}$ **do**
            Select action $a_i^t \sim \pi_i^{\text{eval}}(h_i^t; \theta_i)$ (Equation (12))
         **end for**
        Apply joint action $a^t = (a_1^t, \ldots, a_{|\mathcal{I}|}^t)$
        Receive next state $s^{t+1} \sim \mathcal{P}(s^t, a^t)$, reward $r^t = \mathcal{R}(s^t, a^t)$, and joint observation $o^{t+1} \sim \Omega(s^t, a^t)$
    **end for**

---

---
**Algorithm 3** Training QMIX with EMAX
---

Initialise parameters $\{\theta_i^k\}_{k=1}^K$ of value functions $\{Q_i^k\}_{k=1}^K$ for each agent $i \in \mathcal{I}$
Initialise parameters $\theta_{\mathrm{mix}}$ and $\bar{\theta}_{\mathrm{mix}}$ of the main and target mixing networks
Initialise empty episodic replay buffer $\mathcal{D}$
**for** each episode **do**
    Obtain initial state $s^0$ and joint observation $o^0$
    **for** each step $t = 0, \cdots, T$ **do**
        **for** each agent $i \in \mathcal{I}$ **do**
            Select action $a_i^t \sim \pi_i^{\mathrm{expl}}(h_i^t; \theta_i)$ (Equation (6))
        **end for**
        Apply joint action $a^t = (a_1^t, \ldots, a_{|\mathcal{I}|}^t)$
        Receive next state $s^{t+1} \sim \mathcal{P}(s^t, a^t)$, reward $r^t = \mathcal{R}(s^t, a^t)$, and joint observation $o^{t+1} \sim \Omega(s^t, a^t)$
    **end for**
    Sample bootstrap masks $\{m^k\}_{k=1}^K$ from Bernoulli$(p)$
    Store episode $(s^{0:T}, h^{0:T}, a^{0:T}, r^{0:T}, \{m_k\}_{k=1}^K)$ in $\mathcal{D}$
    **for** each model in the ensemble $k = 1, \ldots, K$ **do**
        Sample batch of episodes $\mathcal{B}$ from $\mathcal{D}$ with $m_k = 1$
        Compute the average loss $\mathcal{L}(\theta^k)$ (Equation (2)) for each timestep $t$ in $\mathcal{B}$ using the QMIX value
decomposition (Equation (10)) and targets (Equation (11))
        Update $\theta^k$ and $\theta_{\mathrm{mix}}$ by minimising $\mathcal{L}(\theta^k)$
    **end for**
    In a set interval, update target mixing network $\bar{\theta}_{\mathrm{mix}} \leftarrow \theta_{\mathrm{mix}}$
**end for**

---

## C.1   Bootstrapped Sampling

As stated in Section 4, to obtain benefits from an ensemble of models, it is important for the models within the ensemble to remain sufficiently different from each other. To maintain diversity across models within the ensemble for EMAX, we train each model on different subsets of the entire experiences collected by using a bootstrapped sampling process. To select bootstrapped samples to train each model in the ensemble, we follow Osband et al. (2016a) and draw a Bernoulli mask $\{m_k\}_{k=1}^K$ for each model in the ensemble whenever an episode is added to the episodic replay buffer. This mask determines whether the $k$-th model within the ensemble is trained on this episode ($m_k = 1$) or not ($m_k = 0$). Each mask is drawn from a Bernoulli distribution with probability $p$ of being 1 and $1 - p$ of being 0, i.e. $m_k \sim$ Bernoulli$(p)$. For $p = 1$, each episode would be used to train each model in the ensemble so all models in the ensemble would receive the same training data. In contrast for a small $p$, the training data is likely to be diverse across models in the ensemble but each model would also only be trained on a small subset of the episodes which might sacrifice learning efficiency. In our experiments, we adopt to use $p = 0.9$ but similar to prior work (Osband et al., 2016a) we have not found the choice of $p$ to significantly affect the performance of our algorithm.

## D   Hyperparameters

For IDQN, VDN, QMIX and extensions with EMAX, we conduct a gridsearch to identify best hyperparameters in one selected task within each environment by evaluating each algorithm configuration for three runs and selecting the hyperparameter configuration which led to highest average evaluation returns throughout training. We largely based our configurations on the reported hyperparameters from Papoudakis et al. (2021) with minimal hyperparameter tuning. Our implementation of IDQN, VDN, QMIX, and EMAX are based on the EPyMARL codebase[5]. For the baseline of MAVEN, CDS, and EMC, we migrated the provided codebase from the authors[6] into EPyMARL to support all environments. For MAVEN, CDS, and EMC, we use the

---

[5]Available at `https://github.com/uoe-agents/epymarl`.
[6]Available at `https://github.com/AnujMahajanOxf/MAVEN`, `https://github.com/lich14/CDS` and `https://github.com/kikojay/EMC`.

hyperparameters identified for QMIX for each environment with the algorithm-specific hyperparameters provided by the authors. For IPPO and MAPPO, we use the best identified hyperparameters reported in Papoudakis et al. (2021).

Table 2: Hyperparameters for IDQN, VDN, QMIX and extensions with EMAX in LBF. The gridsearch was conducted in Foraging-10x10-4p-3f-coop for 4M time steps, and the bold entries corresponding to the best identified configuration.

| Algorithm | Hyperparameter | Value |
|---|---|---|
| Shared | $\gamma$ | 0.99 |
| | Activation function | ReLU |
| | Parameter sharing | True |
| | Optimiser | Adam |
| | Maximum gradient norm | 5 |
| | Minimum $\epsilon$ | 0.05 |
| | Evaluation $\epsilon$ | 0.05 |
| | Learning rate | $e^{-4}$ |
| | Target update frequency | 200 |
| | Replay buffer capacity (episodes) | 5,000 |
| | Batch size (episodes) | 32 |
| QMIX | Mixing embedding size | 32 |
| | Hypernetwork embedding size | 64 |
| IDQN | Network architecture | FC, **FC + GRU** |
| | Network size | 64, **128** |
| | Reward standardisation | False, **True** |
| | $\epsilon$ decay steps | **50,000**, 200,000 |
| VDN | Network architecture | FC, **FC + GRU** |
| | Network size | 64, **128** |
| | Reward standardisation | False, **True** |
| | $\epsilon$ decay steps | 50,000, **200,000** |
| QMIX | Network architecture | **FC**, FC + GRU |
| | Network size | 64, **128** |
| | Reward standardisation | False, **True** |
| | $\epsilon$ decay steps | 50,000, **200,000** |
| IDQN-EMAX | Network architecture | FC, **FC + GRU** |
| | Network size | 64, **128** |
| | Reward standardisation | False, **True** |
| | UCB uncertainty coefficient $\beta$ | 0.1, 0.3, **1** |
| VDN-EMAX | Network architecture | FC, **FC + GRU** |
| | Network size | 64, **128** |
| | Reward standardisation | False, **True** |
| | UCB uncertainty coefficient $\beta$ | **0.1**, 0.3, 1 |
| QMIX-EMAX | Network architecture | FC, **FC + GRU** |
| | Network size | 64, **128** |
| | Reward standardisation | False, **True** |
| | UCB uncertainty coefficient $\beta$ | 0.1, **0.3**, 1 |

Table 3: Hyperparameters for IDQN, VDN, QMIX and extensions with EMAX in BPUSH. The gridsearch was conducted in BPUSH $12 \times 12$ 2ag for 7.5M time steps, and the bold entries corresponding to the best identified configuration.

| Algorithm | Hyperparameter | Value |
|---|---|---|
| Shared | $\gamma$ | 0.99 |
| | Activation function | ReLU |
| | Parameter sharing | True |
| | Optimiser | Adam |
| | Maximum gradient norm | 5 |
| | Minimum $\epsilon$ | 0.05 |
| | Evaluation $\epsilon$ | 0.05 |
| | Learning rate | $e^{-4}$ |
| | Target update frequency | 200 |
| | Replay buffer capacity (episodes) | 5,000 |
| | Batch size (episodes) | 32 |
| QMIX | Mixing embedding size | 32 |
| | Hypernetwork embedding size | 64 |
| IDQN | Network architecture | FC, **FC + GRU** |
| | Network size | 64, **128** |
| | Reward standardisation | False, **True** |
| | $\epsilon$ decay steps | **50,000**, 200,000 |
| VDN | Network architecture | FC, **FC + GRU** |
| | Network size | 64, **128** |
| | Reward standardisation | False, **True** |
| | $\epsilon$ decay steps | 50,000, **200,000** |
| QMIX | Network architecture | **FC**, FC + GRU |
| | Network size | 64, **128** |
| | Reward standardisation | False, **True** |
| | $\epsilon$ decay steps | 50,000, **200,000** |
| IDQN-EMAX | Network architecture | FC, **FC + GRU** |
| | Network size | 64, **128** |
| | Reward standardisation | False, **True** |
| | UCB uncertainty coefficient $\beta$ | 0.1, 0.3, **1** |
| VDN-EMAX | Network architecture | FC, **FC + GRU** |
| | Network size | 64, **128** |
| | Reward standardisation | False, **True** |
| | UCB uncertainty coefficient $\beta$ | **0.1**, 0.3, 1 |
| QMIX-EMAX | Network architecture | FC, **FC + GRU** |
| | Network size | 64, **128** |
| | Reward standardisation | False, **True** |
| | UCB uncertainty coefficient $\beta$ | 0.1, **0.3**, 1 |

Table 4: Hyperparameters for IDQN, VDN, QMIX and extensions with EMAX in RWARE. The gridsearch was conducted in RWARE $11 \times 10$ 4ag for 5M time steps, and the bold entries corresponding to the best identified configuration.

| Algorithm | Hyperparameter | Value |
|---|---|---|
| Shared | $\gamma$ | 0.99 |
| | Activation function | ReLU |
| | Parameter sharing | True |
| | Optimiser | Adam |
| | Maximum gradient norm | 5 |
| | Minimum $\epsilon$ | 0.05 |
| | Evaluation $\epsilon$ | 0.05 |
| | Learning rate | $e^{-4}$ |
| | Target update frequency | 200 |
| | Replay buffer capacity (episodes) | 5,000 |
| | Batch size (episodes) | 32 |
| QMIX | Mixing embedding size | 32 |
| | Hypernetwork embedding size | 64 |
| IDQN | Network architecture | FC, **FC + GRU** |
| | Network size | 64, **128** |
| | Reward standardisation | False, **True** |
| | $\epsilon$ decay steps | **50,000**, 200,000 |
| VDN | Network architecture | FC, **FC + GRU** |
| | Network size | 64, **128** |
| | Reward standardisation | False, **True** |
| | $\epsilon$ decay steps | **50,000**, 200,000 |
| QMIX | Network architecture | **FC**, FC + GRU |
| | Network size | 64, **128** |
| | Reward standardisation | False, **True** |
| | $\epsilon$ decay steps | **50,000**, 200,000 |
| IDQN-EMAX | Network architecture | FC, **FC + GRU** |
| | Network size | 64, **128** |
| | Reward standardisation | False, **True** |
| | UCB uncertainty coefficient $\beta$ | 0.1, **0.3**, 1 |
| VDN-EMAX | Network architecture | FC, **FC + GRU** |
| | Network size | 64, **128** |
| | Reward standardisation | False, **True** |
| | UCB uncertainty coefficient $\beta$ | 0.1, **0.3**, 1 |
| QMIX-EMAX | Network architecture | FC, **FC + GRU** |
| | Network size | 64, **128** |
| | Reward standardisation | False, **True** |
| | UCB uncertainty coefficient $\beta$ | 0.1, **0.3**, 1 |

Table 5: Hyperparameters for IDQN, VDN, QMIX and extensions with EMAX in MPE. The gridsearch was conducted in Spread for 1M time steps, and the bold entries corresponding to the best identified configuration.

| Algorithm | Hyperparameter | Value |
|---|---|---|
| Shared | $\gamma$ | 0.99 |
| | Activation function | ReLU |
| | Parameter sharing | True |
| | Optimiser | Adam |
| | Maximum gradient norm | 5 |
| | Minimum $\epsilon$ | 0.05 |
| | Evaluation $\epsilon$ | 0.05 |
| | Learning rate | $e^{-4}$ |
| | Target update frequency | 200 |
| | Replay buffer capacity (episodes) | 5,000 |
| | Batch size (episodes) | 32 |
| QMIX | Mixing embedding size | 32 |
| | Hypernetwork embedding size | 64 |
| IDQN | Network architecture | **FC**, FC + GRU |
| | Network size | 64, **128** |
| | Reward standardisation | False, **True** |
| | $\epsilon$ decay steps | **50,000**, 200,000 |
| VDN | Network architecture | **FC**, FC + GRU |
| | Network size | 64, **128** |
| | Reward standardisation | False, **True** |
| | $\epsilon$ decay steps | **50,000**, 200,000 |
| QMIX | Network architecture | **FC**, FC + GRU |
| | Network size | 64, **128** |
| | Reward standardisation | False, **True** |
| | $\epsilon$ decay steps | **50,000**, 200,000 |
| IDQN-EMAX | Network architecture | FC, **FC + GRU** |
| | Network size | 64, **128** |
| | Reward standardisation | False, **True** |
| | UCB uncertainty coefficient $\beta$ | 0.1, 0.3, **1** |
| VDN-EMAX | Network architecture | FC, **FC + GRU** |
| | Network size | 64, **128** |
| | Reward standardisation | False, **True** |
| | UCB uncertainty coefficient $\beta$ | **0.1**, 0.3, 1 |
| QMIX-EMAX | Network architecture | FC, **FC + GRU** |
| | Network size | 64, **128** |
| | Reward standardisation | False, **True** |
| | UCB uncertainty coefficient $\beta$ | 0.1, **0.3**, 1 |

# E  Normalised Evaluation Returns

Below, we visualise the normalised evaluation returns across all tasks within each of the four environments. Evaluation returns are normalised between 0 and 1 for each task within the environment with the minimum and maximum achieved evaluation return of any algorithm before computing the interquartile mean and confidence intervals over all tasks and runs.

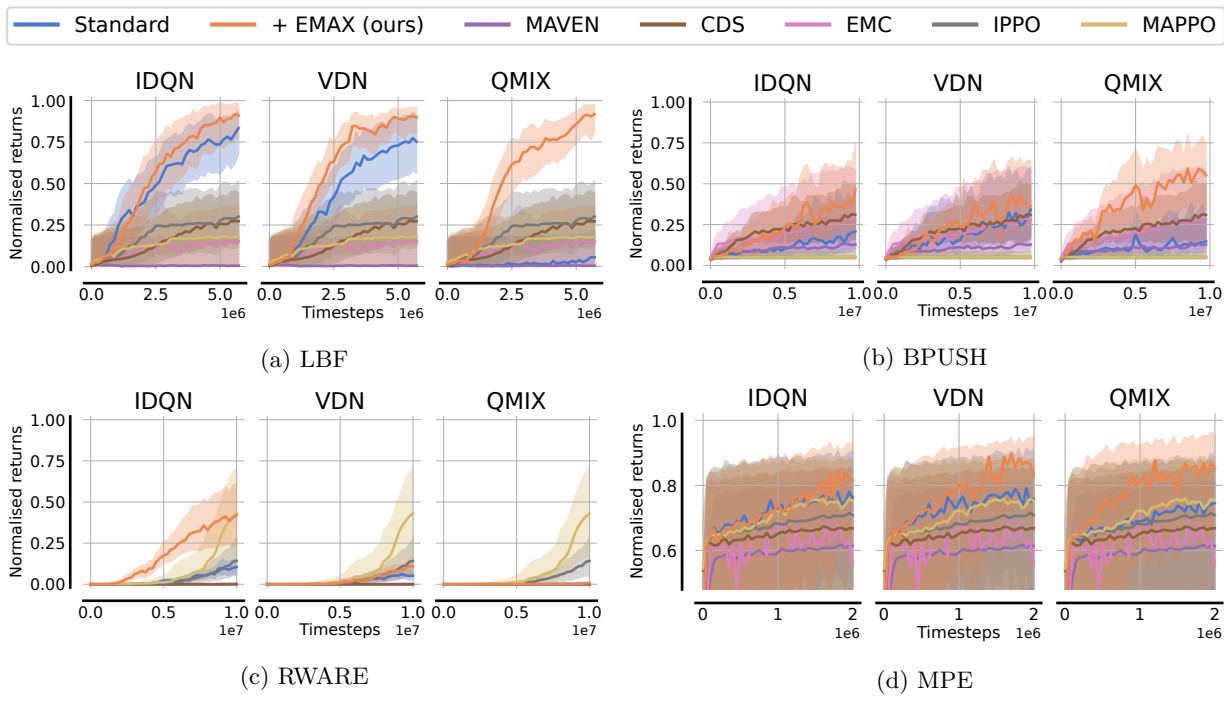

Figure 9: Interquartile mean and 95% confidence intervals of normalised evaluation returns for all algorithms in each environment.

# F   Individual Task Evaluation Returns

## F.1   Level-Based Foraging

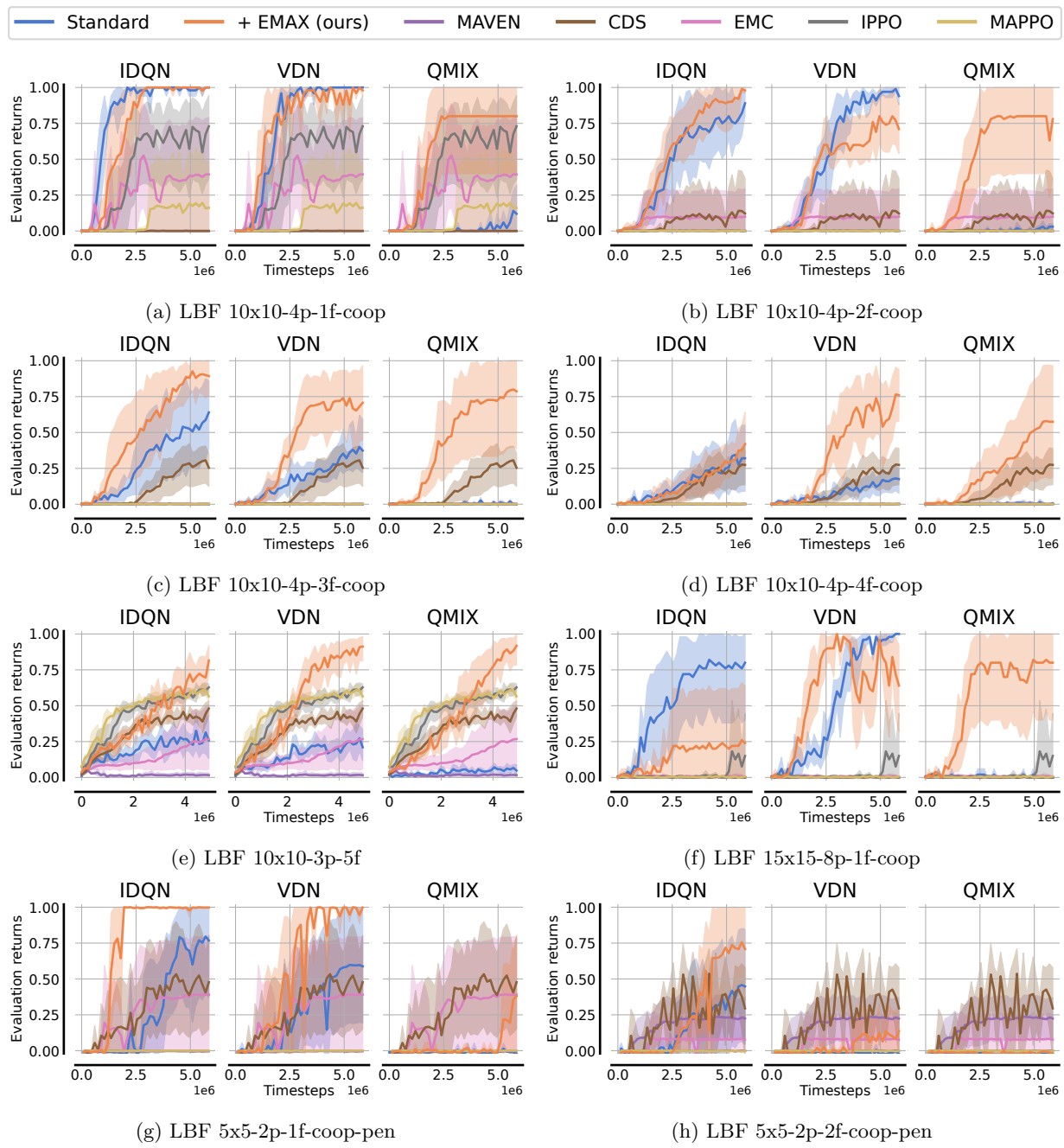

Figure 10: Mean and 95% confidence intervals of evaluation returns for all algorithms in LBF tasks.

## F.2 Boulder-Push

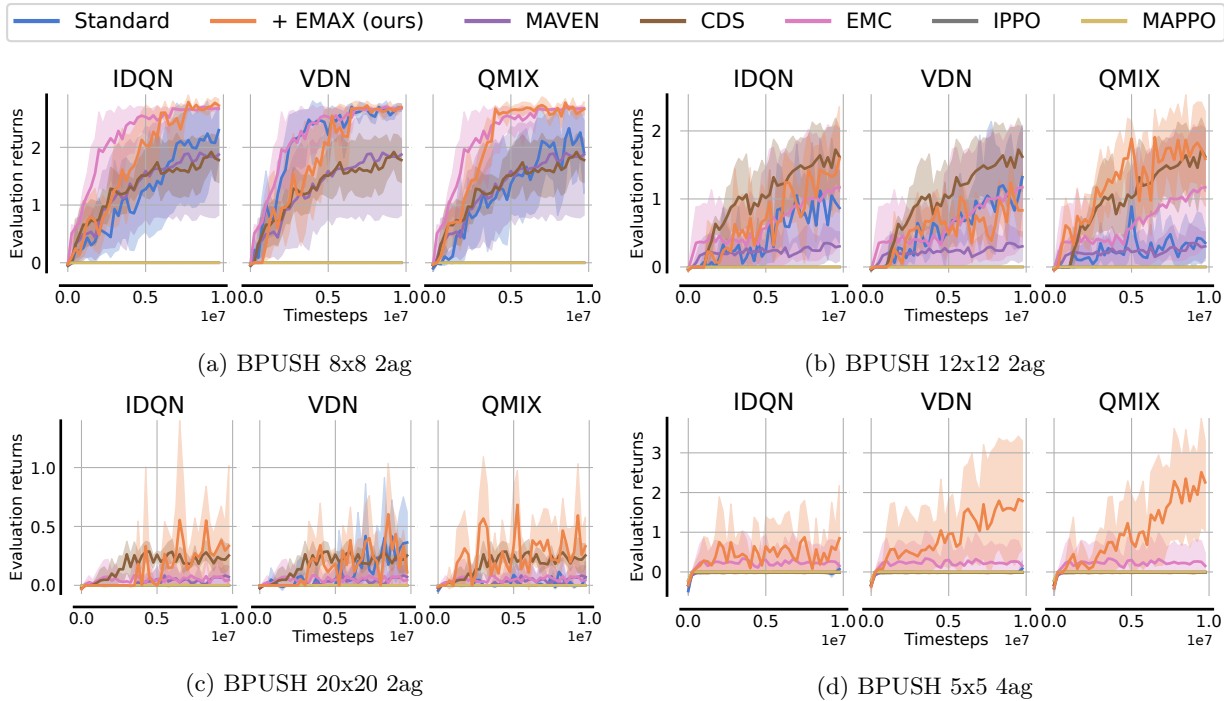

(a) BPUSH 8x8 2ag

(b) BPUSH 12x12 2ag

(c) BPUSH 20x20 2ag

(d) BPUSH 5x5 4ag

Figure 11: Mean and 95% confidence intervals of evaluation returns for all algorithms in BPUSH tasks.

## F.3 Multi-Robot Warehouse

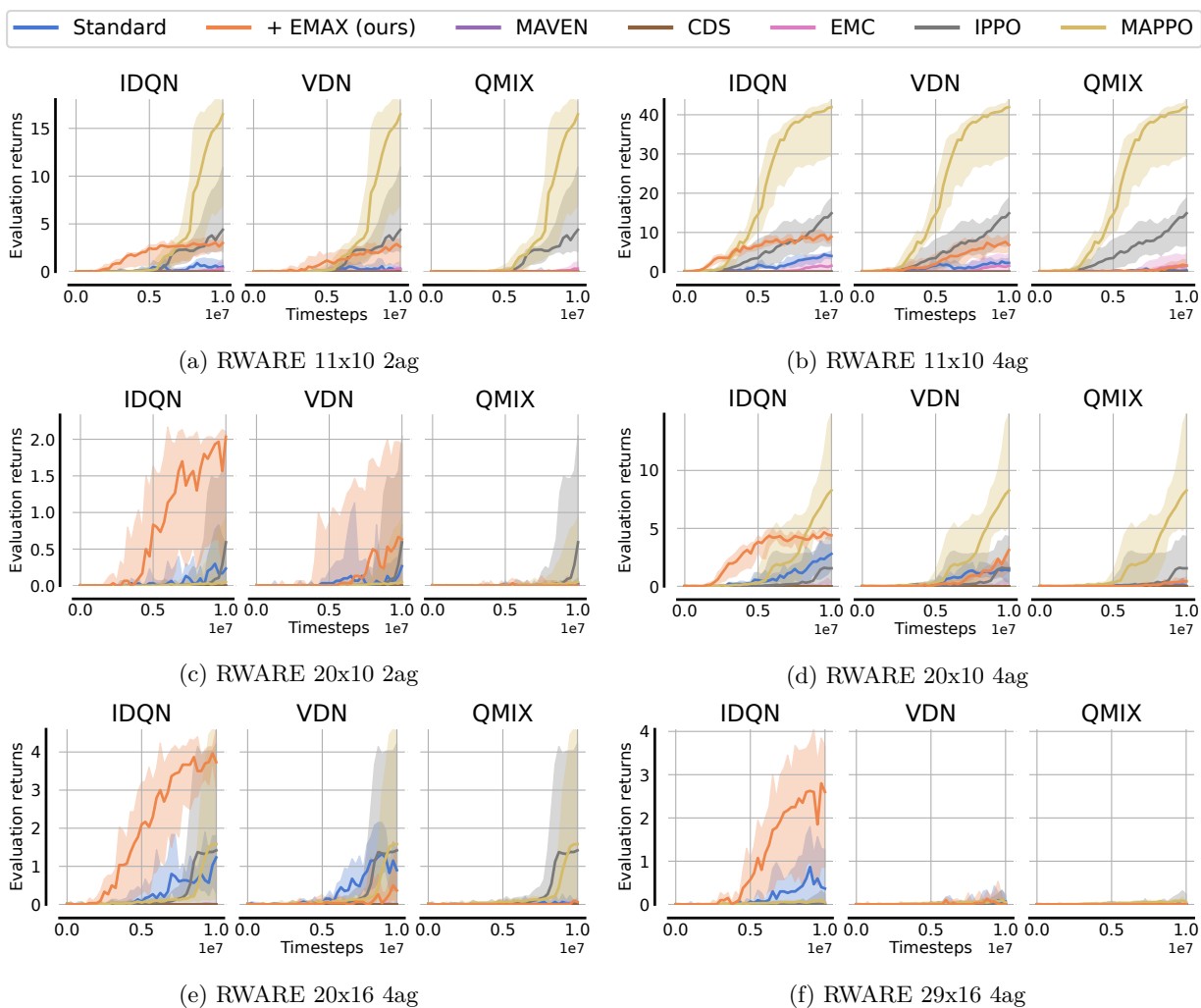

Figure 12: Mean and 95% confidence intervals of evaluation returns for all algorithms in RWARE tasks.

## F.4 Multi-Agent Particle Environment

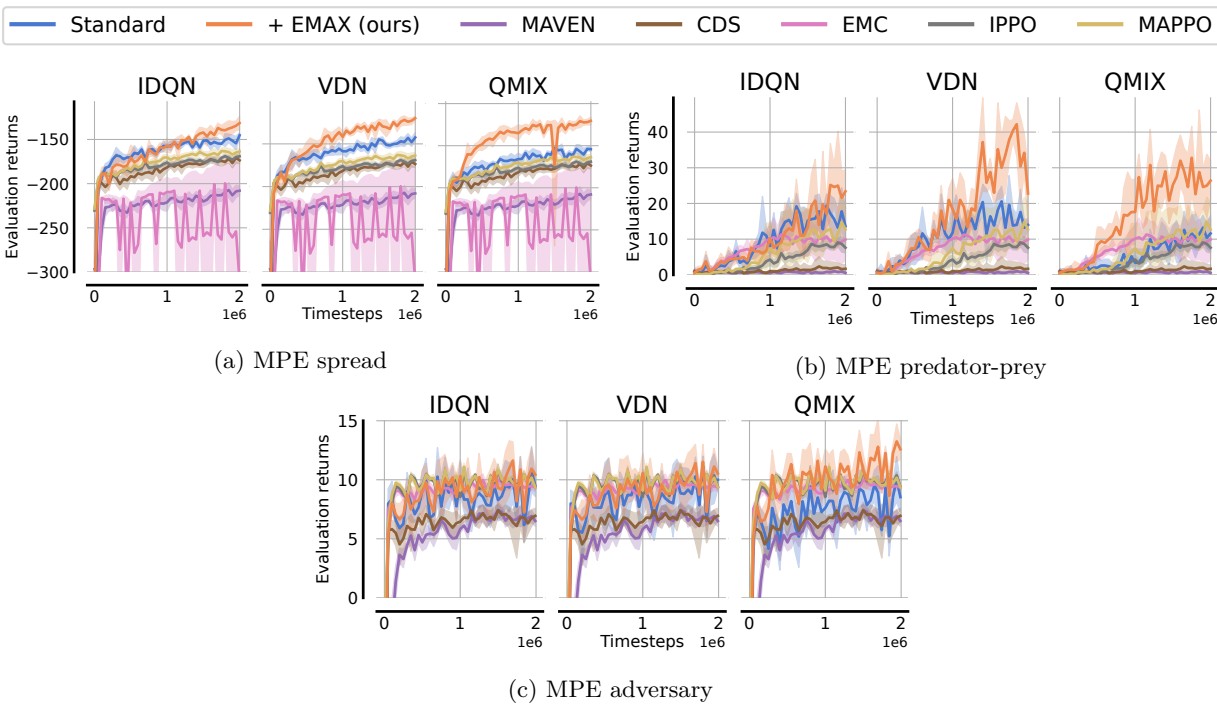

(a) MPE spread

(b) MPE predator-prey

(c) MPE adversary

Figure 13: Mean and 95% confidence intervals of evaluation returns for all algorithms in MPE tasks.

# G   Comparison with Larger Baseline Networks

As seen in Table 1, the computational cost of using ensembles of value functions with EMAX are considerable. To investigate whether the performance of the vanilla algorithms can reach the performance of EMAX at comparable computational cost, we evaluate all vanilla algorithms for larger network sizes. We keep the overall architecture of networks identical, so all value function networks consist of one hidden layer projecting the input observations to a hidden size of $d^h$, followed by a gated recurrent unit (GRU) (Cho et al., 2014) with identical hidden dimensionality, and a final linear layer projecting the hidden output state of the GRU to action-values for each action with the dimensionality of the action space of an individual agent $i$, i.e. $|\mathcal{A}_i|$. We evaluate the vanilla algorithms with hidden sizes of $d^h \in \{128, 256, 512\}$ and compare their performance to EMAX with $K = 5$ models in the ensemble and hidden size of 128. The number of total parameters resulting from these models for one LBF and one RWARE task are shown in Table 6. As we can see, EMAX with $K = 5$ (and hidden size of 128) has exactly five times more parameters in the model compared to the baseline with one model with hidden size of 128. The baseline model with hidden size of 256 is comparable to the model size of EMAX while the baseline model with hidden size of 512 is roughly three times larger than the ensemble of EMAX.

Figure 14 shows the evaluation returns of all vanilla baseline algorithms for varying model sizes compared to EMAX with $K = 5$ models in the ensemble. As we can see, the vanilla algorithms are unable to make effective use of larger networks and reach similar evaluation returns to the original baseline with hidden sizes of 128 despite four times and fifteen times more parameters in the model. Despite these larger networks, the vanilla algorithms are unable to reach the performance of EMAX with $K = 5$ models in the ensemble. This suggests that the ensemble in EMAX is important to make effective use of the increase in parameters and EMAX does not outperform the baselines due to its larger computational budget.

| Task | $|o|$ | $|\mathcal{A}_i|$ | Base (128) | Base (256) | Base (512) | EMAX ($K = 5$) |
|---|---|---|---|---|---|---|
| LBF 10x10-4p-3f-coop | 25 | 6 | 103,174 | 402,950 | 1,592,326 | 515,870 |
| RWARE $11 \times 20$ 4ag | 95 | 5 | 112,005 | 420,613 | 1,627,653 | 560,025 |

Table 6: Observation dimensionality and the resulting number of parameters within the main value function networks for baseline algorithms with hidden sizes of 128, 256, and 512, as well as for EMAX with $K = 5$ models in the ensemble and hidden size of 128 for one LBF and RWARE task.

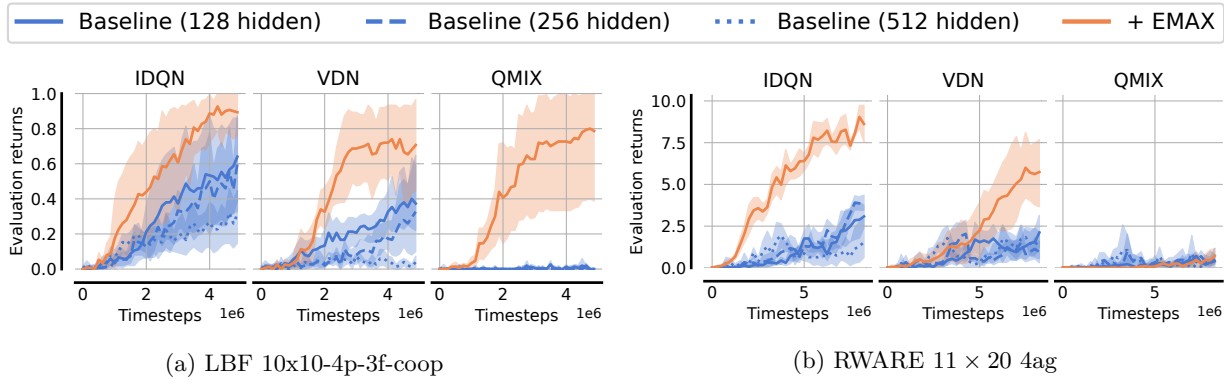

(a) LBF 10x10-4p-3f-coop

(b) RWARE $11 \times 20$ 4ag

Figure 14: Mean and 95% confidence intervals of evaluation returns for all vanilla algorithms with default and larger network sizes, and EMAX extensions.

