# OpenReview forum: "Ensemble Value Functions for Efficient Exploration in Multi-Agent Reinforcement Learning"
_TMLR — Rejected by TMLR_

### Review · Reviewer_9sxj · 2024-04-30

**Summary Of Contributions:**

This paper proposes a novel technique for improving exploration in cooperative multi-agent reinforcement learning (MARL) using an ensemble of value functions (EMAX). Here, the framework leverages uncertainty estimates, target values and majority voting to improve exploration and overall performances. The paper shows that EMAX is a general framework that can be used in conjunction with several value-based algorithms such as DQN, VDN, and QMIX, showing superior empirical performances (as compared to common baselines in the literature) on a wide range of multi-agent tasks in a foraging, warehouse management and MPE environments.

**Audience:**

Yes

**Claims And Evidence:**

No

**Requested Changes:**

Please address my comments in the Weaknesses above.

**Strengths And Weaknesses:**

Strengths:

The problem tackled is important and the idea presented in the paper (though simple) is well-motivated and very intuitive. The experimental results show significant advantages for EMAX, albeit in simple multi-agent
environments. The detailed ablation study in the paper is extremely well-done. These ablation experiments make the advantages of the different components of the algorithm clear.

Weaknesses:

1) The experimental results seem to consider only small grid-world based environments. Given that this is an empirical paper on exploration in cooperative MARL, I would expect to see results on comparatively larger environments such as Google Research Football and Starcraft.

2) The paper has several inconsistencies. First, the navigation example the paper shows in Figure 1 mentions that the "agents receive identical rewards when moving but receive varying rewards when attempting to pick-up". However, the paper later clarifies that it considers the Dec-POMDP setting where all agents should get the same numerical reward at all times. Second, after motivating that each agent learns its own ensemble networks (as evidenced by the network weights, parameterized by the agent index $\theta_i$), the paper says that all agents share the same ensemble networks in the end of Section 4. If all agents share the same networks why does Figure 2 show each agent learning separate Q functions? Also, why is there a summation over the agent indices in Eq.9 if all agents are just learning the same Q function? Third, if the focus of the paper is on partially observable environments (Dec-POMDP
as mentioned in Section 3), why are experiments provided on fully observable environments (as mentioned for MPE). If the environment is fully observable, what is making the setting multi-agent? This seems like a single-agent setting with one centralized agent picking a multi-dimensional action that corresponds to the  actions for each of the other "agents" in each move, since the rewards are the same for all agents. Why should this be treated as a multi-agent environment in the first place?

3) The paper is motivated by the use of uncertainty to drive exploration. However, I am not convinced by the approach of the paper. The paper says that "value estimates will quickly converge in states that require no cooperation and will exhibit high disagreement in states that require cooperation", which may not be particularly true in environments with sparse rewards (as is the focus of the paper). In sparse reward environments, the agents may need to cooperate strongly in some states, while they may not be rewarded for cooperation immediately at that state. Such rewards may come much later in the horizon. In this case, how does the proposed approach learn to explore in such states that require cooperation. All through the paper, it seems like the assumption is that the agents will get immediate rewards upon showing strong cooperation in the states that require cooperation (as discussed in the example in Figure 1).

4) The paper says that each model is trained on bootstrapped samples of the entire experience, however the details are missing. How exactly is the bootstrapping done and what is the motivation behind it?

5) How is independent DQN used with EMAX (i.e., the IDQN-EMAX method)? While the use of EMAX with VDN and QMIX updates are provided in Section 4, the loss function for independent DQN with EMAX is not provided. Also, in independent Q learning, each agent learns its own Q function conditioned on its own state and action, however EMAX just learns one (ensemble of) Q function for all agents. So what exactly is independent in this case?

6) The full pseudo-code of the algorithms that are used in the paper (EMAX + VDN, EMAX + QMIX, EMAX + IDQN) should be present somewhere in the paper (at-least in the appendix). The paper currently does not have these algorithms detailed anywhere in the paper. Will the source code for the experiments be open-sourced? The paper does not seem reproducible in its current state.

---

> ### Author Response · Authors · 2024-05-13
> **Response to Reviewer 9sxj (1/5)**
>
> We thank the reviewer for their comments, and are encouraged to hear they found our method well-motivated and very intuitive, and that they found our empirical ablation very well done. Below, we respond to their comments.
>
> > The experimental results seem to consider only small grid-world based environments. Given that this is an empirical paper on exploration in cooperative MARL, I would expect to see results on comparatively larger environments such as Google Research Football and Starcraft.
>
> We can understand that visually complex environments grounded in games played by humans represent interesting benchmarks for decision making, but we believe and would like to emphasise that the environments considered in our evaluation study are complex coordination and exploration problems that in some cases have been shown to be more challenging to solve for MARL algorithms than GRF and SMAC (e.g. [1, 2]). We believe that further experiments in GRF and SMAC would not provide any additional insights that are not already demonstrated by our extensive evaluation, and would also like to highlight the substantial computational cost that would be involved in such experiments. Lastly, we would like to highlight that several of the evaluation environments have clearly defined states in which strong cooperation of agents is required (e.g. picking-up high-level food in LBF, and pushing the boulder in BPUSH). This allows us to cleanly study and evaluate the main claims of our work and elucidate our main findings.
>
> To further substantiate our justification, we would like to highlight the complexity of existing evaluation environments:
>
> - Multi-robot warehouse: In this environment, agents only receive non-zero rewards when they successfully deliver a requested shelf in the warehouse. Rewards are thus very sparse which makes most MARL algorithm fail to learn to deliver any shelves. EMAX is, to the best of our knowledge, the first value-based MARL algorithm to achieve high returns in this environment and even outperforms previously best-performing [1] on-policy algorithms in IPPO and MAPPO.
> - Level-based foraging: In this environment, agents have to select specific joint actions to achieve positive rewards by collectively picking up food of higher levels. This makes positive rewards sparse and hard to obtain and constitutes a high level of required coordination in these tasks. Identifying such coordination policies between the agents and succeed in these tasks, requires effective exploration and can be challenging in harder tasks with larger grid sizes or degree of cooperation as demonstrated by our evaluation where most baselines fail to learn effective solutions for tasks with large grids, many agents or food that always requires cooperation (marked with "-coop" in the task name).
> - Boulder push: In this environment, all agents have to work together by pushing the boulder at the same time step to be able to obtain any positive rewards. Additionally, any miscoordination in the form of some but not all agents trying to push the boulder leads to negative rewards. These penalties further complicate the discovery and convergence to the successful coordination policy across all agents. We see that all baselines exhibit unstable or no learning in the majority of tasks with none of the baselines achieving positive rewards in a task requiring the cooperation of four agents. In contrast, EMAX significantly stabilises and speeds up learning for smaller BPUSH tasks with two agents and is able to learn to solve the task which requires the cooperation of four agents.
> - Multi-agent particle environment: The MPE environment is different from the other evaluation environments in that it does not represent a discrete grid-world environment but a continuous 2D coordination problem with comparably dense rewards and coordination occurring over many time steps (e.g. agents having to coordinate their movement to cover all landmarks in the cooperative navigation / spread task). This represents a meaningfully different task from RWARE, LBF and BPUSH, that demonstrates that even in this setting EMAX can substantially improve performance of existing value-based MARL algorithms.

---

> ### Author Response · Authors · 2024-05-13
> **Response to Reviewer 9sxj (2/5)**
>
> In contrast, SMAC and GRF feature comparably dense rewards with challenging (temporal and multi-agent) credit assignment since successful cooperation of agents often only leads to large rewards for winning or scoring in SMAC and GRF, respectively, at later timesteps. However, it is not clear what exploration challenges SMAC and GRF provide that go beyond the environments considered in our evaluation. Additionally, the challenge of temporal and multi-agent credit assignment is already present in our evaluation environments. In RWARE and MPE tasks, agents are not immediately rewarded for successful cooperation (e.g. in the form of avoiding collisions and effectively spreading of agents) but such reward is only received with a temporal delay (in RWARE) or such cooperation only leads to overall higher episodic returns without being particularly apparent at any individual timestep (in MPE). For multi-agent credit assignment, agents need to identify their contribution to received rewards of successful food collection in LBF and shelf deliveries in RWARE, just to give two examples of how this challenge is present in our environments.
>
> By comparing learning curves in existing work which evaluate in both SMAC and LBF [1, 2], we can see that value-based MARL algorithms (such as IDQN/ IQL and QMIX) often require more timesteps to converge to good solutions in LBF compared to SMAC tasks or do not manage to reach such solutions at all in LBF. However, as established by our evaluation, EMAX significantly boosts the performance of these value-based MARL algorithms across many diverse environments which we believe is a major demonstration of the value of our approach.
>
> In short, we do not believe that the environments in our evaluation, albeit visually less complex than GRF and SMAC, are simple in comparison as MARL benchmarks but represent hard problems for these algorithms (as demonstrated by the performance of baseline algorithms) which is why we studied the impact of our EMAX framework in such environments.
>
> [1] Papoudakis, Georgios, Filippos Christianos, Lukas Schäfer, and Stefano V. Albrecht. "Benchmarking multi-agent deep reinforcement learning algorithms in cooperative tasks." Advances in Neural Information Processing Systems, Datasets and Benchmarks Track (2021).
>
> [2] Mguni, David Henry, Haojun Chen, Taher Jafferjee, Jianhong Wang, Longfei Yue, Xidong Feng, Stephen Marcus Mcaleer, Feifei Tong, Jun Wang, and Yaodong Yang. "MANSA: learning fast and slow in multi-agent systems." In International Conference on Machine Learning, pp. 24631-24658. PMLR, 2023.
>
> > the navigation example the paper shows in Figure 1 mentions that the "agents receive identical rewards when moving but receive varying rewards when attempting to pick-up". However, the paper later clarifies that it considers the Dec-POMDP setting where all agents should get the same numerical reward at all times.
>
> We meant to express that any agent receives the same reward of zero whenever choosing to move independent of the actions of other agents (since without its collaboration no reward can be received for picking-up the object), but the received reward when choosing to pick-up in the state where both agents are next to the goal object varies depending on whether the other agent also decides to pick-up. We modified this paragraph to make this more clear in the paper.
>
> > after motivating that each agent learns its own ensemble networks (as evidenced by the network weights, parameterized by the agent index), the paper says that all agents share the same ensemble networks in the end of Section 4. If all agents share the same networks why does Figure 2 show each agent learning separate Q functions? Also, why is there a summation over the agent indices in Eq.9 if all agents are just learning the same Q function?
>
> We would like to clarify that the ensemble model receives an agent ID as a one-hot encoded vector as input in addition to the observation history and outputs the action-values for each action of an agent for each value function within the ensemble. The agent ID input allows the model to learn specialised value functions for different agents where necessary. Since the agent IDs are different across agents and the observation histories of agents may differ from each other, we need to compute the output of the ensemble model for the inputs and actions of all agents in Eq.9 even when sharing parameters across agents. This is why there remains a summation over the agent indices in Eq. 9 and why we decided to visualise multiple Q-functions in Figure 2. To make this more clear, we now make the design choice of agent ID inputs explicit in Section 5.1: "For every algorithm and task, agents share network parameters with networks receiving the agent identity in the form of a onehot vector as additional input."

---

> ### Author Response · Authors · 2024-05-13
> **Response to Reviewer 9sxj (3/5)**
>
> Additionally, we consider the choice of sharing parameters across agent models as one way of implementing EMAX but not a core part of our method. EMAX could equally be applied without sharing parameters across agents. Therefore, we present EMAX in Section 4 with the most general notation in which the network of agent $i$ has parameters $\theta_i$. Parameter sharing simply follows from this notation with $\forall i, j \in \mathcal{I}: \theta_i = \theta_j$. Lastly, we note that, as stated in the paragraph on "ensemble value functions" in our work, sharing parameters across models of multiple agents is common practise in multi-agent reinforcement learning and has often been shown to improve learning efficiency in cooperative settings (e.g. [1, 3, 4]), and is particularly desirable in our work since we train larger more computationally costly models.
>
> [3] Rashid, Tabish, Mikayel Samvelyan, Christian Schroeder De Witt, Gregory Farquhar, Jakob Foerster, and Shimon Whiteson. "Monotonic value function factorisation for deep multi-agent reinforcement learning." Journal of Machine Learning Research 21, no. 178 (2020): 1-51.
>
> [4] Mahajan, Anuj, Tabish Rashid, Mikayel Samvelyan, and Shimon Whiteson. "Maven: Multi-agent variational exploration." Advances in neural information processing systems 32 (2019).
>
> > if the focus of the paper is on partially observable environments (Dec-POMDP as mentioned in Section 3), why are experiments provided on fully observable environments (as mentioned for MPE).
>
> As stated in Section 5.1 and Appendix B, all RWARE tasks are partially observable since agents only observe shelves and other agents in their close proximity so our evaluation does include six partially observable tasks.
>
> > If the environment is fully observable, what is making the setting multi-agent? This seems like a single-agent setting with one centralized agent picking a multi-dimensional action that corresponds to the actions for each of the other "agents" in each move, since the rewards are the same for all agents. Why should this be treated as a multi-agent environment in the first place?
>
> We agree with the reviewer that in a fully observable environment it is possible to model the problem as a single-agent problem by learning a single centralised controller that selects joint actions across all agents in the task. However, such an approach is rarely desirable since the joint action space grows exponentially with the number of agents. For example in a LBF task with 4 agents with each agent selecting one of six possible actions, the joint action space contains $6^4=1,296$ possible joint actions. For 8 agents, this grows to $6^8=1,679,616$ joint actions. Learning a policy over such a large action space is very challenging if not infeasible. By modelling the decision making problem as a multi-agent problem, we effectively decompose the large joint action space into much more manageable individual action spaces of all agents to simplify learning. We further note that there exist many fully observable environments that are commonly studied in MARL, e.g. Google Research Football (GRF), which is most commonly treated as a fully observable problem, the majority of tasks within the multi-agent particule environment (MPE), and a variety of board games such as Go and chess which have a long history in the multi-agent decision making community.

---

> ### Author Response · Authors · 2024-05-13
> **Response to 9sxj (4/5)**
>
> > The paper says that "value estimates will quickly converge in states that require no cooperation and will exhibit high disagreement in states that require cooperation", which may not be particularly true in environments with sparse rewards (as is the focus of the paper). In sparse reward environments, the agents may need to cooperate strongly in some states, while they may not be rewarded for cooperation immediately at that state. Such rewards may come much later in the horizon. In this case, how does the proposed approach learn to explore in such states that require cooperation. All through the paper, it seems like the assumption is that the agents will get immediate rewards upon showing strong cooperation in the states that require cooperation (as discussed in the example in Figure 1).
>
> While the motivational example in Figure 1 has immediate sparse rewards following successful cooperation, as correctly stated by the reviewer, we would like to point out that several environments considered in our evaluation do not follow that pattern. For example in the MPE task of spread, agents have to cooperate by avoiding collisions with each other and moving to different landmarks but agents receive no immediate or sudden reward for better cooperation. In RWARE, agents need to cooperate to avoid moving to identical requested shelves and blocking each others paths but no rewards is received by agents for successfully avoiding such conflicts. Agents only receive rewards for successfully delivering shelves so over time they can learn that they need to avoid such conflicts to maximise their returns. Given that EMAX is still effective in these environments, we conclude that EMAX does not assume or rely on the fact that rewards are immediately received following cooperation.
>
> To see why EMAX might still be effective in exploring in these tasks, we would like to highlight that the exploration policy of EMAX follows actions that result in high average episodic returns and large variation of episodic returns. Since returns are computed over longer time horizons (as long as the episode continues), these metrics can capture the potential of cooperation as resulting in varying returns given by variability of future rewards and not just immediate rewards. However, due to discounting applied during the computation of episodic returns, EMAX will prefer actions that lead to higher variation of rewards in the near future over actions that lead to such variation in the futher future.
>
> We hope this helps in clarifying when EMAX can be effective as supported by our experiments. That being said, we agree with the reviewer that the majority of cooperative benchmarks considered our evaluation and in general require multiple agents to cooperate and immediately reward agents for successful cooperation which is also why we decided to follow this simple pattern in our illustrative example.
>
> > The paper says that each model is trained on bootstrapped samples of the entire experience, however the details are missing. How exactly is the bootstrapping done and what is the motivation behind it?
>
> The motivation behind training each model within the ensemble on bootstrapped samples of all experiences is to ensure diversity of the models in the ensemble. If all models in the ensemble would be trained with the exact same training data, then they might quickly collapse to the same function and, thus, the ensemble would lose its effect.
>
> We implement this idea in the same way as prior work using ensemble models for reinforcement learning (e.g. [5, 6]) by drawing a Bernoulli mask ${m_k}$ for each model $k=1,\ldots,K$ in the ensemble whenever an episode is added to the episodic replay buffer. This mask determines whether this episode is used to train the $k$-th model within the ensemble ($m_k = 1$) or not ($m_k = 0$). Each mask is drawn from a Bernoulli distribution with probability $p$ of being $1$ and $1-p$ of being $0$, i.e. $m_k \sim \text{Bernoulli}(p)$. For $p=1$, each episode would be used to train each model in the ensemble and all models in the ensemble would receive the same training data. In contrast for small $p$, the training data would likely be highly diverse across models in the ensemble but each model would also only see a small subset of the training data which might sacrifice performance. In our experiments, we adopt to use $p=0.9$ but similar to prior work [5] we have not found the choice of $p$ to significantly affect the performance of our algorithm. We added these details in Appendix C and refer to them in the main paper when mentioning the bootstrapped sampling process.

---

> > ### Author Response · Authors · 2024-05-13
> > **Response to 9sxj (5/5)**
> >
> > [5] Osband, Ian, Charles Blundell, Alexander Pritzel, and Benjamin Van Roy. "Deep exploration via bootstrapped DQN." Advances in neural information processing systems 29 (2016).
> >
> > [6] Lee, Kimin, Michael Laskin, Aravind Srinivas, and Pieter Abbeel. "Sunrise: A simple unified framework for ensemble learning in deep reinforcement learning." In International Conference on Machine Learning, pp. 6131-6141. PMLR, 2021.
> >
> > > How is independent DQN used with EMAX (i.e., the IDQN-EMAX method)? While the use of EMAX with VDN and QMIX updates are provided in Section 4, the loss function for independent DQN with EMAX is not provided. Also, in independent Q learning, each agent learns its own Q function conditioned on its own state and action, however EMAX just learns one (ensemble of) Q function for all agents. So what exactly is independent in this case?
> >
> > The loss for IDQN with EMAX is given in Equation 8. We clarified the text before the equation to provide more context for the equation.
> >
> > Regarding the comment about independent DQN and parameter sharing, independent learning is generally used for MARL algorithms which apply single-agent RL solutions to each agent, i.e. each agent in the multi-agent setting is learning independently using RL. This still holds in EMAX. As stated above, we consider parameter sharing an implementation detail and refer to prior work that show its impact on learning in fully cooperative settings [1, 3, 4]. Furthermore, we can consider the ensemble model with the agent ID input of agent $i$ as a separate Q-function from the functions of other agents when providing a different agent ID input.
> >
> > > The full pseudo-code of the algorithms that are used in the paper (EMAX + VDN, EMAX + QMIX, EMAX + IDQN) should be present somewhere in the paper (at-least in the appendix). The paper currently does not have these algorithms detailed anywhere in the paper. Will the source code for the experiments be open-sourced? The paper does not seem reproducible in its current state.
> >
> > We added full pseudocode for training IDQN + EMAX and QMIX + EMAX in Appendix C of the paper as well as pseudocode for evaluating any algorithm with EMAX. The pseudocode of VDN + EMAX is almost identical to QMIX + EMAX, so we simply describe how this differs from the provided pseudocode for QMIX + EMAX. Unfortunately we are unable to release the source-code publicly since this research project resulted from an industry collaboration which does not permit an open-source release. However, we are hopeful that the details added in Appendix C will provide sufficient information to reproduce our research.

---

> > > ### Comment · Reviewer_9sxj · 2024-05-26
> > > **Reply to the author responses.**
> > >
> > > I thank the authors for their responses. I acknowledge that some of my concerns have been addressed. The changes made by the authors and their responses does improve my understanding. Broadly, I think the paper continues to have the following shortcomings which must be still acknowledged:
> > >
> > > 1) The experiments with MPE seem unnecessary in the paper and only increases confusion. This environment is fully-observable, but the paper focuses on partially-observable settings. The authors' explanation that these environments were included since they have a long history in the multi-agent decision-making community seems very primitive. There are several environments with long histories in the MARL community, and we cannot expect research works to use all those environments. One must only use those test-beds that aligns with the settings that are considered in the paper. Further, a fully observable setting with full cooperation (i.e., all agents sharing the same reward) is just a single-agent setting with a central controller picking actions for all agents, i.e., there is nothing multi-agent in this formulation. The authors mention that the joint action space is large for a central agent, however, when each individual agent (learning MARL policies) selects actions, it has to reason over the joint action space of all agents anyway (unless if the agents are independent, the paper considers several non-independent algorithms), so I see no advantages to formulating this as a multi-agent problem. This is the classic curse of dimensionality in MARL, and that does not change while having a central controller or using individual agents deciding actions. In GRF and MPE, previous works have considered mixed motive settings, where multiple agents across teams get different rewards. Having different rewards in a fully observable environment necessitates a multi-agent treatment. However, in this work agents' rewards are the same for all agents and there is no need for a multi-agent treatment for this environment.
> > >
> > > 2) Including the agent ID seems like a very hand wavy way of resolving the issue of all agents sharing the same ensemble network. In all the experiments considered in the paper, the agents in all environments seem to be interchangeable. I do not believe having the agent ID makes any difference to the learning of the agents in the experimental domains considered in the paper. The paper requires more experimentation in heterogeneous domains where some agents have a different action space, for example.
> > >
> > > 3) Given the premise of the paper, the paper should have demonstrated EMAX on at-least one environment where agents have to cooperate early in the horizon, but only gets sparse rewards very late in the horizon (especially given the fact that the paper motivates itself using sparse reward environments). Based on the evidence in the paper, I do not think it can be conclusively claimed that EMAX will be helpful in such environments at all.
> > >
> > > 4) Unfortunately, based on the information that is available in the paper, I do not feel confident that the paper can be fully reproduced. Even the details about the random seeds used is not included in the paper. Given that the authors mentioned that the source code will not be open-source, I feel that the paper has only a limited possibility of being reproducible.
> > >
> > > Given the above reasons, I am voting for a rejection. Nonetheless, I do agree that the paper has its merits, and I am open to be convinced about other strong reasons to accept the paper (in-spite of its limitations) by either the authors or the other reviewers.

---

> ### Author Response · Authors · 2024-06-07
> **Reply to Reviewer 9sxj (1/3)**
>
> We thank the reviewer for their response and are glad to hear that our response have improved their understanding of our work. Below we aim to further address the remaining concerns:
>
> > The experiments with MPE seem unnecessary in the paper and only increases confusion. This environment is fully-observable, but the paper focuses on partially-observable settings.
>
> We would like to clarify that our work is not focused on the problem of partially observable settings. We formulate problems under the Dec-POMDP formalism for generality (since all fully observable cooperative problems can be formulated as Dec-POMDPs) and evaluate in both fully observable (e.g. BPUSH and considered LBF tasks) and partially observable (e.g. RWARE) environments to demonstrate the broad applicability of our approach. However, our approach is not tailored to partially observable settings.
>
> Additionally, we would like to mention that the MPE environment provides agents with egocentric observations, which, although they pertain all information of the environmental state, are different for all agents. We further decided to include the MPE environment since it is meaningfully different from other included environment since it features dense rewards, continuous observations, and the predator prey and adversary MPE tasks include significant stochasticity in the transition function by the fact that the adversary agent in these tasks is controlled by a stochastic pre-trained policy.
>
> > The authors' explanation that these environments were included since they have a long history in the multi-agent decision-making community seems very primitive. There are several environments with long histories in the MARL community, and we cannot expect research works to use all those environments. One must only use those test-beds that aligns with the settings that are considered in the paper.
>
> We agree with the reviewer and would like to clarify that we did focus our evaluation on environments that fall under our considered setting -- fully-cooperative multi-agent problems with a particular focus on hard exploration problems that require cooperation of multiple agents to solve. As explained in Section 5.1, Appendix B, and in more detail in our initial reponse to the reviewer's questions, all our used multi-agent environments represent such hard multi-agent cooperation problems. Beyond this focus, our evaluation considers a wide range of tasks and environments which vary in their further properties (but all fall into the category of hard multi-agent cooperation problems) to demonstrate the generality of our EMAX approach. Examples for such diversity covered by our evaluation are fully observable (e.g. BPUSH) and partially observable environments (e.g. RWARE), sparse reward (e.g. RWARE and LBF) and dense reward environments (e.g. MPE), and discrete (e.g. LBF, RWARE) and continuous observations/ states (e.g. MPE). To be clear, we do not argue that EMAX can be effective in any setting but our evaluation does demonstrate the efficacy of EMAX in a diverse set of tasks which all fall in the category of hard multi-agent cooperation problems.

---

> ### Author Response · Authors · 2024-06-07
> **Reply to Reviewer 9sxj (2/3)**
>
> > Further, a fully observable setting with full cooperation (i.e., all agents sharing the same reward) is just a single-agent setting with a central controller picking actions for all agents, i.e., there is nothing multi-agent in this formulation. The authors mention that the joint action space is large for a central agent, however, when each individual agent (learning MARL policies) selects actions, it has to reason over the joint action space of all agents anyway (unless if the agents are independent, the paper considers several non-independent algorithms), so I see no advantages to formulating this as a multi-agent problem.
>
> We would like to clarify that the mentioned central learning approach, i.e. learning a single policy over the joint action space with centralised execution, and most MARL approach, which typically assume decentralised execution, are distinct in the policy that is being learned. A central learning approach learns a single policy over the joint action space. In contrast, most MARL approaches assume the necessity of decentralised execution and, thus, learn multiple policies with each policy defining a distribution over the action space of an individual agent which is often significantly smaller as the joint action space (as outlined by the example in our original response to the reviewer). Independent learning approaches fall into this paradigm of decentralised execution, but also centralised training decentralised execution (CTDE) algorithms like VDN and QMIX follow this paradigm. In that sense, the central learning approach needs to directly reason over the large joint action space when learning a policy, while decentralised execution approaches only indirectly reason over the actions of other agents through (potentially centralised) value functions and the data distribution of experiences being dependent on the policies of all agents. Modelling a problem as a decentralised execution MARL problem in this way can simplify the challenge of learning the policies compared to a central learning approach, but introduces other challenges such as the multi-agent credit assignment problem and additional non-stationarity of the learning process from each agent's perspective. We do not argue that either approach is preferrable, but would like to demonstarte that our MARL approach is equally valid and can be desirable since its decomposition can simplify learning.
>
> > Including the agent ID seems like a very hand wavy way of resolving the issue of all agents sharing the same ensemble network. [...] The paper requires more experimentation in heterogeneous domains where some agents have a different action space, for example.
>
> We do not make any statements about the efficacy of our approach in heterogeneous domains and therefore have not done any experiments in these settings.
>
> > Given the premise of the paper, the paper should have demonstrated EMAX on at-least one environment where agents have to cooperate early in the horizon, but only gets sparse rewards very late in the horizon (especially given the fact that the paper motivates itself using sparse reward environments). Based on the evidence in the paper, I do not think it can be conclusively claimed that EMAX will be helpful in such environments at all.
>
> We would like to reiterate that multiple tasks considered in our evaluation feature sparse temporally delayed rewards for successful cooperation. As stated in our original reply, the RWARE environment has highly sparse rewards [1, 2, 3] and requires agents to cooperate to avoid collisions and blocking each others path but agents are only rewarded for delivering requested shelves. These deliveries often occur dozens of timesteps after successful cooperation to resolve conflicts. This property of temporally delayed rewards combined with highly sparse rewards make RWARE a hard multi-agent cooperation problem and were our motivation for including it in our evaluation. Furthermore, we would like to highlight that EMAX improves performance of IDQN and VDN in all six RWARE tasks, leading on average to 330% and 252% higher final evaluation returns compared to the vanilla algorithms, respectively.

---

> ### Author Response · Authors · 2024-06-07
> **Reply to Reviewer 9sxj (3/3)**
>
> To visualise and demonstrate that collisions and agents blocking each others' path does indeed occur and delay the successful delivery of requested shelves, we generated rollouts of several episodes using a checkpoint of IDQN-EMAX agents after 4M timesteps of training in the RWARE 11x10 4ag task. These rollouts as well as a edited video with annotations can be accessed through this anonymous repository: https://anonymous.4open.science/r/emax_rebuttal_data-B2D8/rware_demonstration_rollouts/annotated_emax_rware_demo.mp4. To further explain these rollouts, red markers indicate the agents with green boxes marking requested shelves. Agents only receive a reward when moving with a green requested shelf onto one of the goal locations at the bottom of the warehouse. This behaviour requires the agents to move towards a requested shelf in the warehouse, selecting the "load" action, and navigating to a goal location. To be able to pick-up new shelves and deliver further requested shelves to obtain more rewards, agents need to then place any shelves after deliveries in an empty spot in the warehouse which can contain shelves. In all of these rollout videos, we can observe agents being blocked in their path by other agents standing in their way which delays their ability to deliver requested shelves for several timesteps. Once the blocking agent successfully cooperates and moves out of the way, shelves can be successfully delivered to receive rewards. We further highlight that not just the rewards for the initial delivery after resolving such conflicts are enabled by the successful cooperation of moving out of the way, but any rewards received by the initially blocked agent through delivering more requested shelves later in the episode are also made possible by resolving the conflict. Such further deliveries can occur hundreds of timesteps later in the episode since each episode runs for 500 timesteps.
>
> As demonstrated, RWARE does feature temporally delayed sparse rewards, as requested by the reviewer, so we believe our evaluation in RWARE already satisfies the request and indicates that the efficacy of EMAX does not rely on rewards being received immediately following cooperation. We leave further investigation into the temporal credit assignment problem, which is not the focus of our work, to future work.
>
> [1] Georgios Papoudakis, Filippos Christianos, Lukas Schäfer, & Stefano V. Albrecht. "Benchmarking Multi-Agent Deep Reinforcement Learning Algorithms in Cooperative Tasks." Advances in neural information processing systems, Track on Datasets and Benchmarks, 2021.
>
> [2] Christianos, Filippos, Lukas Schäfer, and Stefano Albrecht. "Shared experience actor-critic for multi-agent reinforcement learning." Advances in neural information processing systems, 2020.
>
> [3] Christianos, Filippos, Georgios Papoudakis, Muhammad A. Rahman, and Stefano V. Albrecht. "Scaling multi-agent reinforcement learning with selective parameter sharing." In International Conference on Machine Learning, 2021.
>
> Lastly, we would like to emphasise the focus of the TMLR journal on technical correctness. We consider our work highly suitable to this venue since it presents technically sound results with strong empirical evidence.

---

### Review · Reviewer_ZqwK · 2024-05-03

**Summary Of Contributions:**

In this work, the authors introduce EMAX, an ensemble-based modification that can be applied to a wide range of MARL algorithms, using ensemble-variance to construct a UCB-like policy.

They provide in a number of domains using a number of base algorithms, and show generally positive results in comparison to the unaltered base algorithms. The authors also provide some exploration of the parameters: large ensembles don't seem to be necessary, and some ablation experiments suggest both the ensemble policy and the choice of a single target network are important for performance.

**Audience:**

Yes

**Claims And Evidence:**

No

**Requested Changes:**

I think the broader changes noted above need to be made, or otherwise addressed in some way: have a resource-fair comparison, the motivating argument needs to be strengthened / modified / removed, and address stochastic outcomes in environments.


Other comments:

The choice of a majority action rule for the policy would seem to require discrete action, which should be noted in the paper.

"In cooperative MARL, agents should focus their exploration on states and action that require cooperation to achieve high value."
They should focus on states that achieve high value, and that will often require cooperation. But not always: if agent A can just force a win for all with no cooperation, that's a fantastic policy.

A nit which is possibly only personal choice. I'm not sure interquartile mean is an appropriate measure for 5 data points, regardless of any use in previous papers. The high and low "outlier" would seem to be as likely to be part of the expected distribution as they are to be actual outliers which should be disregarded, given this little data.

Add something answering the question "what cooperation is required?" when describing the multi-robot warehouse.

Multi-agent particle environment: "The agents receive rewards by moving an agent to the close landmark"  goal landmark?

Evaluation results: "These results mostly arise from improved sample efficiency for EMAX extensions". Include a justification for this statement, including the use of "mostly"  (how much? what else?)

"EMAX with K=8 performs identical or worse"  The "or worse" seems worth some explanation

**Strengths And Weaknesses:**

One strength of this work is that the proposed method (modification?) is reasonably uncomplicated, and could be implemented with fairly low effort for some impressive improvements in some domains.

Considering weaknesses, one small point is that the paper does not quite compare like to like when comparing against the baselines. The modification does require both more memory and compute, just as a single larger network and/or longer training runs would. I very much believe that increasing the resources for the baseline methods would not have them matching the performance of EMAX in the experiments, but ~2 the compute and 5 times the model size is a non-trivial difference.

Another point is that I'm not convinced by the author's argument that EMAX drives exploration to states where cooperation is required, making it particularly suited to MARL. With a UCB-like policy, EMAX will explore higher variance states more. Past that, there seem to me to be a couple of problems. (i) States which require cooperation can also be low variance, in "all actors required" states: unless cooperation is already likely enough that all actors happen to cooperate by chance, the expected value will be low, with low variance. (ii) The choice of a single target network, which works better in practice in the experiments, seems to be at odds with having variance across values in the ensemble in areas where cooperation is still possible but not yet the most likely policy choice. (iii) This argument doesn't seem to mix well with a stochastic environment, where the highest variance states might be there is randomness in the environment (and could have low value), rather than being states which are opportunities for cooperation.
None of this changes the experimental results, which are positive. I do think that argument should be changed in the presentation, and I think it might be worth using an environment that has non-trivial variance in expected return from stochastic environment behavior.

---

> ### Author Response · Authors · 2024-05-13
> **Response to Reviewer ZqwK (1/2)**
>
> We thank the reviewer for their comments. We appreciate that they found our method reasonably uncomplicated and acknowledged the impressive improvements it can bring. Below, we address the reviewer's comments:
>
> > The modification does require both more memory and compute, just as a single larger network and/or longer training runs would. I very much believe that increasing the resources for the baseline methods would not have them matching the performance of EMAX in the experiments, but ~2 the compute and 5 times the model size is a non-trivial difference.
>
> We thank the reviewers for their suggestion. We added experiments comparing the performance of EMAX with $K=5$ and a hidden size of $128$ to vanilla algorithms with hidden sizes of $128$ (original), $256$ (4x model size), and $512$ (16x model size) with identical architecture in one LBF and one RWARE task. We show the results in Appendix G of the revised paper. The results support our hypothesis and the hypothesis of the reviewer that EMAX still outperforms the baselines, even if they are given a comparable or larger computational budget
>
>
> > [...] not convinced by the author's argument that EMAX drives exploration to states where cooperation is required, making it particularly suited to MARL. With a UCB-like policy, EMAX will explore higher variance states more. Past that, there seem to me to be a couple of problems.
>
> We thank the reviewer for outlining their concerns with the intuitive explanation of the EMAX exploration strategy. Below, we will discuss each concern.
>
> >(i) States which require cooperation can also be low variance, in "all actors required" states: unless cooperation is already likely enough that all actors happen to cooperate by chance, the expected value will be low, with low variance.
>
> We agree with the reviewer that states which require cooperation can lead to low variance across value estimates in the ensemble if the agents are already consistently succeeding at the cooperation. This is likely to occur in settings where the optimal joint action is easily attainable. In this case, no further exploration is required so instead we might say that the EMAX exploration policy drives exploration to states where cooperation is required and not yet consistently achieved. However, we note that this case is unlikely in settings where the optimal joint action is difficult to identify which are the settings the exploration policy of EMAX is designed for.
>
> >(ii) The choice of a single target network, which works better in practice in the experiments, seems to be at odds with having variance across values in the ensemble in areas where cooperation is still possible but not yet the most likely policy choice.
>
> We are unsure which experiments the reviewer refers regarding the "single target network". In EMAX, there are no target networks since target values are computed as aggregate values across the ensemble of value functions instead of using target networks. In Figure 8, we show an ablation, where we substitute the target computation of EMAX with target networks, performs significantly worse than EMAX in two tasks. Would the reviewer be able to elaborate so we can hopefully address their concerns?
>
> > (iii) This argument doesn't seem to mix well with a stochastic environment, where the highest variance states might be there is randomness in the environment (and could have low value), rather than being states which are opportunities for cooperation. None of this changes the experimental results, which are positive. [...] I think it might be worth using an environment that has non-trivial variance in expected return from stochastic environment behavior.
>
> We note that several environments considered in our evaluation feature stochasticity. All tasks feature significant stochasticity in the initial state distribution (by randomising various elements of the state), and MPE predator prey and adversary additionally feature stochastic transitions by the fact that the adversary agent of these tasks is controlled by a stochastic pre-trained policy. Despite this pre-trained policy causing stochastic outcomes of each transition, we still see that EMAX improves or matches the performance of each baseline in these tasks.
>
> That being said, we agree with the reviewer's assessment that stochasticity in the environment's transition or reward function can contribute to high variance in returns and, thus, influence the EMAX exploration policy. However, our experiments demonstrate that EMAX is not overly sensitive to stochastic environments and can still work effectively under such conditions.
>
> > The choice of a majority action rule for the policy would seem to require discrete action, which should be noted in the paper.
>
> Thank you for pointing out this assumption. It is correct that EMAX assumes discrete action spaces (just as the extended base algorithms of DQN, VDN, and QMIX). We now state this assumption in footnote 1 in Section 3.

---

> > ### Comment · Reviewer_ZqwK · 2024-05-14
> > **clarification re. target values**
> >
> > > > (ii) The choice of a single target network, which works better in practice in the experiments, seems to be at odds with having variance across values in the ensemble in areas where cooperation is still possible but not yet the most likely policy choice.
> > >
> > > We are unsure which experiments the reviewer refers regarding the "single target network". In EMAX, there are no target networks since target values are computed as aggregate values across the ensemble of value functions instead of using target networks. In Figure 8, we show an ablation, where we substitute the target computation of EMAX with target networks, performs significantly worse than EMAX in two tasks. Would the reviewer be able to elaborate so we can hopefully address their concerns?
> >
> > I did mean the choice of having a single set of target values for all ensemble members. Having shared targets seems to be at odds with having diversity across the ensemble.

---

> > > ### Author Response · Authors · 2024-05-15
> > > **Response to Reviewer Clarification**
> > >
> > > We would like to thank the reviewer for clarifying their question.
> > >
> > > > I did mean the choice of having a single set of target values for all ensemble members. Having shared targets seems to be at odds with having diversity across the ensemble.
> > >
> > > The reviewer is correct in that the target values for all members of the ensemble are computed identically (average across the value estimates of all members of the ensemble). However, this target computation does not prevent the models in the ensemble to learn slightly different value functions which is sufficient for EMAX.
> > >
> > > One contributor as to why the value functions within the ensemble learn slightly different functions, even if target computation is identical, is that all value functions are computed on slightly different data by training each model on a bootstrapped subset of the overall training data. This process is described in more detail in the new Appendix C, and we would also like to refer the reviewer to our Response to Reviewer SJgH (1/2) in which we illustrate how this process helps to obtain diverse value estimates across the ensemble irrespective of the target computation at the example of a simple matrix game. Besides the difference in training data, we also see that different initialisation of all models across the ensemble helps to ensure diversity of the models in the ensemble.
> > >
> > > The diversity in value estimates across the ensemble can also be seen in Figure 6d (added during our revisions) which shows the mean and standard deviation of value estimates across the ensemble for different types of actions in states where agents can cooperate by choosing the picking-up action. This plot clearly shows that there is a notable deviation in value estimates across the ensemble (as indicated by the standard deviation), and that such deviation is larger for the pick-up action with the potential for cooperation. The latter is more clearly illustrated by this new plot (requested by reviewer SJgH) which only visualises the standard deviation of action-values across the ensemble and shows notable deviation in value estimates across the ensemble all throughout training until learning curves are close to convergence: https://anonymous.4open.science/r/emax_rebuttal_data-B2D8/q_values_coop_stds.pdf
> > >
> > > These results indicate that our approach is sufficient to ensure that models within EMAX remain diverse until convergence despite identical target computation.

---

> > > > ### Comment · Reviewer_ZqwK · 2024-05-18
> > > >
> > > > Thank you to the authors for the changes: I think they do improve the paper.
> > > >
> > > > I'm glad that switching away from IQM doesn't change any of the plots in any substantial way, and there's no longer any concern that significant variance in a configuration doesn't show up by classifying one of the few runs as an outlier.
> > > >
> > > > It's also good to directly see in appendix G that the improvements can not be simply replicated by using a baseline with more parameters.
> > > > There are were two possibilities in the "larger network and/or longer training runs" comment, and I still think there is a (smaller) outstanding question of EMAX performance compared to baseline with extra computation in the form of more training: the choice of the x-axis being steps rather than time. In a bunch of the scenarios this is clearly answered, simply because the baseline seems to have plateaued at a level far below EMAX. However, some of the other baseline runs appear to still be improving, so that being roughly twice as fast as EMAX per step  and effectively changing the slope by a factor of 2, could be non-trivial.
> > > >
> > > >
> > > > I do understand that the variance in the ensemble value estimates will reflect the variance in the sampled observations, but that includes an increasing amount of data over time: if you ignored NN training, as in your example, the variance in the value estimates across the ensemble is going to be roughly the same magnitude as the squared error of the value estimates. Non-trivial variance with 5 steps, but a lot less with a lot more data. This is maybe reasonable for a UCB estimate of a stationary distribution, but it seems worth some care distinguishing there being diversity, and there being decreasing diversity but it not being a problem. Another issue is that the values are not necessarily stationary, as multiple agent behaviors can cycle. Or another: how much does that relationship still hold with NN training and Adam? The plots in 6d show ensemble variance decreasing as expected early, but it seems to plateau -- which is convenient -- and it's a bit hard to tell from the plots and the width of the overlapping shaded regions, but variance also seems to be similar in magnitude for actions -- contrary to the text, and the cooperation variance argument. If there are specific timesteps that the authors are looking at, it might be better to have that variance information in a table, so it's not hidden by overlapping regions of a graph.
> > > >
> > > > Or the short version: as originally stated, I think the empirical results are good, but I don't agree (or at least still remain unconvinced by) the motivating story or the arguments about why things are working. In my opinion, the paper would be further improved by noting the authors' inspiration about cooperation and variance, and focus on the results and avoid trying to also empirically validate the cooperation variance inspiration.

---

> > > > > ### Author Response · Authors · 2024-05-18
> > > > > **Reply to Reviewer ZqwK**
> > > > >
> > > > > We thank the reviewer for engaging in discussion with us and helping us improve the clarity of our work, their comments are very helpful and we are glad to heard that they find many of their previous concerns addressed.
> > > > >
> > > > > > I still think there is a (smaller) outstanding question of EMAX performance compared to baseline with extra computation in the form of more training: the choice of the x-axis being steps rather than time. In a bunch of the scenarios this is clearly answered, simply because the baseline seems to have plateaued at a level far below EMAX. However, some of the other baseline runs appear to still be improving, so that being roughly twice as fast as EMAX per step and effectively changing the slope by a factor of 2, could be non-trivial.
> > > > >
> > > > > We agree with the reviewer that some baselines might reach similar performance to EMAX in some of the tasks given identical training (wall-clock) time (i.e. using more timesteps in the environment), but we would also argue that the focus of our work was on demonstrating improved performance and in particular sample efficiency in hard cooperative exploration problems, which our evaluation clearly shows, and we are not making any claims regarding improved wall-clock efficiency of EMAX. As mentioned by the reviewer, it appears that in many tasks we find that baselines seem to converge to lower performance to EMAX, indicating that even with further training time these might not reach similar performance to EMAX, which is an encouraging result, but we do not consider such time-efficiency a core motivation of our work.
> > > > >
> > > > > > The plots in 6d show ensemble variance decreasing as expected early, but it seems to plateau -- which is convenient -- and it's a bit hard to tell from the plots and the width of the overlapping shaded regions, but variance also seems to be similar in magnitude for actions -- contrary to the text, and the cooperation variance argument.
> > > > >
> > > > > We agree that it is slightly difficult to tell whether the variance/ standard deviation of different actions across the ensemble value estimates in Figure 6d are similar or not. To address this concern we would like to point the reviewer to a new plot (requested by reviewer SJgH): https://anonymous.4open.science/r/emax_rebuttal_data-B2D8/q_values_coop_stds.pdf
> > > > >
> > > > > Similar to Figure 6d, the plot shows the standard deviation of value estimates across the ensemble for different types of action (no-op, movement, and pick-up) in states where agents can cooperate by choosing the pick-up action but it only shows the standard deviation as lines. This plot clearly shows that the magnitude of deviations of action values across the ensemble for the pick-up action (which has the potential for cooperation in these states we show action-value estimates for) are meaningfully larger than the respective deviations for all the other (non-cooperative) actions. Additionally, we can read two further points from inspecting this plot alongside Figure 6a and 6d:
> > > > > 1. The standard deviation across the ensemble for all types of actions is similar early in training but the gap between the deviation for the cooperative pick-up action and other actions slowly increases as agents start to sometimes receive reward for attempting to pick-up (when successfully cooperating with other agents) and sometimes do not (when failing to cooperate with other agents).
> > > > > 2. When inspecting the deviation of action values for and the performance of algorithms shown in Figure 6a, we can see that once agents successfully cooperate most of the time, the standard deviation for the pick-up action starts to reduce. For QMIX with EMAX, we can see that this reduction ends in the standard deviation of action values for the cooperative pick-up action and non-cooperative actions reaching similar levels once close-to-optimal performance is reached since now agents almost always cooperative successfully. This further indicates that, as desired, EMAX incentivizes exploration of cooperative actions as long as such cooperation is not reliably achieved yet, but such bias towards cooperative actions diminishes as the policy starts to reliably cooperate successfully.
> > > > >
> > > > >
> > > > > > In my opinion, the paper would be further improved by noting the authors' inspiration about cooperation and variance, and focus on the results and avoid trying to also empirically validate the cooperation variance inspiration.
> > > > >
> > > > > Does the reviewer find the argument and results outlined above convincing? If not, would the reviewer be able to state what remains unclear and how we might be able to address their concerns? Again, we greatly appreciate the reviewer’s efforts and engagement in discussion and their help in further strengthening our work!

---

> ### Author Response · Authors · 2024-05-13
> **Response to Reviewer ZqwK (2/2)**
>
> > "In cooperative MARL, agents should focus their exploration on states and action that require cooperation to achieve high value." They should focus on states that achieve high value, and that will often require cooperation. But not always: if agent A can just force a win for all with no cooperation, that's a fantastic policy.
>
> We agree with the reviewer that cooperation is only desirable where necessary and updated the referenced sentence as follows: "In multi-agent problems which require agents to cooperate to achieve high returns, agents should focus their exploration on states and actions that require cooperation." That being said, most cooperative MARL problems are (intentionally) setup such that agents need to cooperate to achieve high returns, which is the motivation behind EMAX encouraging agents to explore such cooperative behaviour. If it is found to be suboptimal, EMAX will eventually deviate and take the non-cooperative optimal action instead.
>
> > A nit which is possibly only personal choice. I'm not sure interquartile mean is an appropriate measure for 5 data points, regardless of any use in previous papers. The high and low "outlier" would seem to be as likely to be part of the expected distribution as they are to be actual outliers which should be disregarded, given this little data.
>
> We thank the reviewer for raising this. We originally decided to use the interquartile mean for the aggregated learning curves across multiple tasks since then we have $5 \cdot \text{number of tasks}$ data points, but for individual tasks agree that merely 5 data points might be insufficient. We revised the learning curves in Section 5.3 and Appendix E to now show average evaluation returns and 95\% confidence intervals. We note that this change did not affect the overall comparison of algorithms and their relative orderings in tasks are preserved.
>
> > Add something answering the question "what cooperation is required?" when describing the multi-robot warehouse.
>
> We added more details about the required cooperation in the multi-robot warehouse environment in Appendix B. To summarise, agents are capable of delivering shelves without interaction with other agents, but agents need to cooperate to avoid blocking each others path in narrow parts of the warehouse and trying to move to and load identical requested shelves. To maximise episodic returns, agents need to learn to avoid such conflicts with other agents which requires them to learn conventions and cooperate.
>
> > Multi-agent particle environment: "The agents receive rewards by moving an agent to the close landmark" goal landmark?
>
> We thank the reviewer for this comment. We clarified the description of the adversary task in Appendix B (and more briefly in Section 5.1) as follows: "Both agents are rewarded for one of them being close to the goal landmark but they are negatively rewarded for the adversary agent moving close to the goal location. Therefore, agents are incentivised to cover both landmarks in order to maximise their rewards since then they are receiving rewards for covering the goal landmark but also preventing the adversary agent from identifying which landmark is the goal landmark."
>
> > Evaluation results: "These results mostly arise from improved sample efficiency for EMAX extensions". Include a justification for this statement, including the use of "mostly" (how much? what else?)
>
> Generally, it is difficult to identify why EMAX extensions reached higher final evaluation returns than the vanilla algorithms. Our evaluation clearly shows that EMAX improves sample efficiency in many tasks but it is possible that with longer training some of the vanilla algorithms would eventually reach similar returns in some tasks. Therefore, we decided to slightly modify the statement to highlight the benefits of EMAX that are further established in our analysis: "These results arise from EMAX improving the sample efficiency and learning stability of the vanilla algorithms, as we will show in Section 5.3."
>
> > "EMAX with K=8 performs identical or worse" The "or worse" seems worth some explanation
>
> As stated shortly after in the same paragraph, "we hypothesise that larger ensemble value functions may require more data to train, thus leading to diminishing benefits for ensembles of many value functions."

---

### Review · Reviewer_SJgH · 2024-05-10

**Summary Of Contributions:**

The authors propose learning ensemble value functions for cooperative multi-agent RL (MARL). They claim several benefits of this ensemble approach: 1) It enables a UCB style exploration policy that focuses on parts of the environments that require agents' cooperation; 2) Improved training stability; 3) At evaluation time, majority voting from the ensembles reduces chance of selecting sub-optimal actions.  This method can be integrated with existing MARL algorithms and demonstrates improved performance on several environments that require cooperation.

**Audience:**

Yes

**Broader Impact Concerns:**

No concerns.

**Claims And Evidence:**

No

**Requested Changes:**

I would like to see plots (or other evidence) showing standard deviation of value estimates indicates the state-action require cooperation. If no such evidence can be provided, the authors should consider modifying their claims on the mechanism of the exploration policy.

**Strengths And Weaknesses:**

Strength:
1. Experiment results are strong, showing clear advantage of the proposed method.
2. Writing is clear, well organized, and is easy to follow.
3. Ablation studies provide useful insights on the impact of various design choices.

Weakness:
1. The authors claim the exploration policy eq (6) "guides agents to explore actions that are promising (as measured
by the mean value estimate) and are likely to require cooperation (as measured by the standard deviation of value estimates)". I find the argument following that statement is quite hand-wavy. The author's essential reasoning is: in states requiring cooperation, "... received rewards for a given action of agent i will vary significantly, since the reward depends on the actions of other agent ..." I agree the received rewards can have large variations, but I fail to see how this necessarily lead to variations _across the ensembles_. On the other hand, variations across ensembles may be due to other factors (such as different initializations). It would be much more convincing if the authors include plots showing state-action pair requiring cooperations indeed has larger variation across ensembles. Otherwise, the observed benefit may be simply due to leveraging uncertainty in value estimate, instead of encouraging cooperation as claimed.

---

> ### Author Response · Authors · 2024-05-13
> **Response to Reviewer SJgH (1/2)**
>
> We thank the reviewer for their comments and are encouraged to hear they found our paper clearly written and experiments strong. Below, we address their key questions.
>
> > The authors claim the exploration policy eq (6) "guides agents to explore actions that are promising (as measured by the mean value estimate) and are likely to require cooperation (as measured by the standard deviation of value estimates)". I find the argument following that statement is quite hand-wavy. The author's essential reasoning is: in states requiring cooperation, "... received rewards for a given action of agent i will vary significantly, since the reward depends on the actions of other agent ..." I agree the received rewards can have large variations, but I fail to see how this necessarily lead to variations across the ensembles. On the other hand, variations across ensembles may be due to other factors (such as different initializations).
>
> To see why variation of rewards following successful or failed cooperation results in variation of value estimates across the ensemble, we would like to point the reviewer to the final paragraph in Section 4 in which we highlight several steps we follow to ensure that models within the ensemble remain diverse. One of these steps is training each model within the ensemble on a different bootstrapped subset of the overall training data. We describe this process in more detail in the new Appendix C.
>
> To illustrate how this process results in value estimates across the ensemble to be different in cases where cooperation is required, let's consider a simple common-reward matrix game with two agents where cooperation is important when selecting action A but no cooperation is required for action B of any agent:
>
> | $R(a_1, a_2)$ | A | B |
> | -------- | -------- | -------- |
> | **A**     | 10     | 1     |
> | **B**     | 1     | 1     |
>
> For simplicity, we will furthermore just look at agent 1 (the same reasoning follows for agent 2) training an ensemble of 3 action-value functions $\{Q^k_1(a_1)\}_{k=1}^3$ which take its own actions as input and try to estimate the expected returns. Agents collect experiences by randomly sampling actions, and each value function within the ensemble is trained on a different subset of experiences as per the bootstrapped sampling process defined in Appendix C. For example for five interactions, we might obtain the following experiences:
> 1. E1: $a_1$ = A, $a_2$ = B --> $r = 1$
> 2. E2: $a_1$ = B, $a_2$ = B --> $r = 1$
> 3. E3: $a_1$ = A, $a_2$ = A --> $r = 10$
> 4. E4: $a_1$ = B, $a_2$ = A --> $r = 1$
> 5. E5: $a_1$ = A, $a_2$ = B --> $r = 1$
>
> Let's say, the 3 ensemble value functions of agent 1 are trained on the following subset of experiences:
> 1. Model: experiences E1, E2, E3, E4
> 2. Model: experiences E2, E4, E5
> 3. Model: experiences: E1, E2, E4, E5
>
> Following these experiences, the action-value functions of Model 1, Model 2, and Model 3 will provide the following expected estimates for the actions of agent 1 (we simply compute the average rewards over all experiences within the training data of a particular model for which $a_1 = \text{A}$ or $a_1 = \text{B}$):
>
> 1. Model:
>     - $Q_1^1(\text{A}) = \frac{1 + 10}{2} = 5$
>     - $Q_1^1(\text{B}) = \frac{1 + 1}{2} = 1$
> 2. Model:
>     - $Q_1^2(\text{A}) = \frac{1}{1} = 1$
>     - $Q_1^2(\text{B}) = \frac{1 + 1}{2} = 1$
> 3. Model:
>     - $Q_1^3(\text{A}) = \frac{1 + 1}{2} = 1$
>     - $Q_1^3(\text{B}) = \frac{1 + 1}{2} = 1$
>
> Following this process, we obtain models which significantly vary in their action-value estimates for action A but no such variation can be observed for action B. This variation arises since Models 2 and 3 have not been trained on any experience for action A in which both agents successfully cooperated to achieve reward 10, but Model 1 has been trained on one such sample. In contrast, all experiences for action B necessarily lead to reward of 1 so all of the models estimate the action value of action B to be 1.
>
> The same process also applies to more complicated sequential decision making problems as targeted in our work. We hope this illustration clarifies how training each model within the ensemble on a different subset of training experiences results in variations of value estimates across the ensemble for actions (and states) which require cooperation but no such variation is expected for actions (and states) that do not require cooperation.

---

> > ### Author Response · Authors · 2024-05-13
> > **Response to Reviewer SJgH (2/2)**
> >
> > > I would like to see plots (or other evidence) showing standard deviation of value estimates indicates the state-action require cooperation. If no such evidence can be provided, the authors should consider modifying their claims on the mechanism of the exploration policy.
> >
> > We thank the reviewer for their suggestion. We added a corresponding plot as Figure 6d in the revised paper. The plot shows the mean and standard deviation of value estimates across the ensemble for different types of action (no-op, movement, and pick-up) in states where agents can cooperate by choosing the pick-up action. As the plot visualises, the standard deviation for the pick-up action is larger than for other actions, indicating the potential of this action to lead to cooperation.

---

> > > ### Comment · Reviewer_SJgH · 2024-05-14
> > > **Minor ask**
> > >
> > > Thanks for the new plot. It shows _some_ evidence of greater variance for state-action pairs requiring co-op. However, I find it challenging to read the magnitude of the standard deviation as it is shown as shaded bands. The plot would be more convincing if the standard deviations are plotted as curves.

---

> > > > ### Author Response · Authors · 2024-05-15
> > > > **Reply to Minor Ask**
> > > >
> > > > We thank the reviewer for their quick response and are encouraged to hear their main concerns on the mechanism of our approach is resolved.
> > > >
> > > > We generated a plot as requested by the reviewer which only visualises the standard deviation of action-values across the ensemble which is shown as shading in Figure 6d. The plot is available through this anonymous repo: https://anonymous.4open.science/r/emax_rebuttal_data-B2D8/q_values_coop_stds.pdf
> > > >
> > > > Similar to Figure 6d, the plot shows the standard deviation of value estimates across the ensemble for different types of action (no-op, movement, and pick-up) in states where agents can cooperate by choosing the pick-up action. As the plot visualises, the standard deviation for the pick-up action is larger than for other actions, indicating the potential of this action to lead to cooperation. Furthermore, we can see that the standard deviation across the ensemble for all types of actions is similar early in training but slowly the gap between the deviation for the cooperative pick-up action and other actions increases as agents sometimes receive reward for attempting to pick-up (when successfully cooperating with other agents) and sometimes do not (when failing to cooperate with other agents). The plot also shows that once agents successfully cooperate most of the time such cooperative states are reached, the standard deviation for the pick-up action slowly reduces, and falls to similar levels as the other actions in QMIX-EMAX once close-to-optimal performance is reached (agents almost always cooperate successfully) at the end of training.

---

> > ### Comment · Reviewer_SJgH · 2024-05-14
> > **resolved**
> >
> > Thanks for the clarification and the updated paper. My main concern on the mechanism is resolved.

---

### Author Response · Authors · 2024-05-13
**Common Response**

We thank all the reviewers for their helpful comments, and are encouraged that they found our evaluation strong and our approach intuitive.

To address the comments of all reviewers, we slightly revised the paper (all revisions are highlighted in the PDF in blue) with the following changes:

### Evaluation
- Following a suggestion of reviewer ZqwK, we added Appendix G which shows experiments comparing the performance of EMAX with $K=5$ and a hidden size of $128$ to vanilla algorithms with hidden sizes of $128$ (original), $256$ (4x model size), and $512$ (16x model size) with identical architecture in one LBF and one RWARE task. The results support our hypothesis and the hypothesis of the reviewer that EMAX still outperforms the baselines, even if they are given a comparable or larger computational budget.
- As suggested by reviewer SJgH, we added a visualisation of the mean and standard deviation of action-value estimates across the ensemble of EMAX for a LBF task (Figure 6d). This plot shows that the standard deviation and mean value estimates correspond to the potential of an action to lead to cooperation across agents.
- Reviewer ZqwK raised concerns about the suitability of the interquartile mean to aggregate performance metrics when the aggregation is only done over five seeds. We agree with these concerns and decided to update all learning curves for individual tasks to show the mean evaluation returns instead of the interquartile mean. We note that this change did not affect the overall comparison of algorithms and their relative orderings in tasks are preserved. For normalised evaluation plots and performance profiles aggregated across several tasks, we still use the interquartile mean.


### Clarifications
- We added Appendix C with pseudocode for EMAX training and evaluation, and a section to provide details on the bootstrapped sampling process used to ensure each model within the ensemble is trained on different training data.
- We clarified that shared networks receive the agent identity in the form of a onehot vector as input in additon to the observation history of an agent, so networks can learn specialised functions for different agents where necessary. We added a sentence stating this design choice in Section 5.1.
- We added footnote 1 in Section 3 to make our assumption of discrete action spaces explicit.
- We clarified the cooperation required by agents in the multi-robot warehouse environment in Appendix B.
- We clarified the reward function and required cooperation in the adversary task of the multi-agent particle environment in Appendix B and Section 5.1.

---

### Public Comment · ~Haque_Ishfaq1 · 2024-05-13
**Related Work on Uncertainty for exploration in RL**

Hi authors,

This looks like an exciting paper. We just wanted to point out few related work which are relevant to your Section 2: Uncertainty for exploration in RL.

1. Ishfaq, Haque, et al. "Provable and Practical: Efficient Exploration in Reinforcement Learning via Langevin Monte Carlo." The Twelfth International Conference on Learning Representations. 2024.

2. Ishfaq, Haque, et al. "Randomized exploration in reinforcement learning with general value function approximation." International Conference on Machine Learning. PMLR, 2021.

Thanks.

---

### Decision · Action_Editor_hJeW · 2024-06-09

**Recommendation:** Reject

**Comment:**

During the discussion, the authors responded to concerns about the connection between uncertainty estimation and improved coordination by pointing to the diversity of environments and improved performance. While I appreciate the diversity of environments considered, this does _not_ prove that states with high value estimation uncertainty are always states where coordination is necessary. For this to be convincing, the authors should either provide theoretical evidence and/or empirical evaluations that go deeper than simply comparing performance curves.

Thus, although I think this work holds merit and could be of interest to the MARL community, I agree with all the reviewers that in its current for it is not yet ready for publication. Specifically, the authors should either substantially tone down the claims of "uncertainty in value estimates implies needed coordination", or provide stronger evidence for this claim.

**Audience:**

This paper would be of interest to the MARL community.

**Claims And Evidence:**

The authors have done a good job at addressing most of the reviewers concerns, and their revised draft reflects this.

I went through all the reviews and author discussion, as well as through the paper myself. At a high-level, it seems to me there are two main contributions of this work:
1) Establishing the connection between value estimation uncertainty and states where coordination is required
2) Introducing a plug-and-play method for helping exploration in MARL environments.

The paper provides good evidence for point #2, and this is something where all reviewers (and myself) are in agreement.

However, one of the main concerns raised by all reviewers is that the central claim and motivation of the paper is not correct; namely: "High variability of rewards implies necessary coordination" (e.g. point #1 above). While the algorithm (and positive empirical results) doesn't necessarily depend on this statement being true, the statement is present throughout the first 4 sections of the paper. To quote from section 4: " Due to this variability of rewards (or lack thereof), value estimates across the ensemble will quickly converge in states that require no or limited cooperation, and will exhibit high disagreement in states that require cooperation. Therefore, the EMAX exploration policy focuses the exploration of agents on state-action pairs that require cooperation." This may be true in the environments considered in this work (which seem to be specifically geared towards cooperative MARL), but is _not_ true in general.

In the abstract, the authors state three main components for EMAX (**bold** added by me): " (1) EMAX uses the uncertainty of value estimates across the ensemble in a UCB policy to guide the exploration. **This exploration policy focuses on parts of the environment which require cooperation across agents and, thus, enables agents to more efficiently learn how to cooperate.** (2) During the optimisation, EMAX computes target values as average value estimates across the ensemble. These targets exhibit lower variance compared to commonly applied target networks, leading to significant benefits in MARL which commonly suffers from high variance caused by the exploration and non-stationary policies of other agents. (3) During evaluation, EMAX selects actions following a majority vote across the ensemble, which reduces the likelihood of selecting sub-optimal actions."

I agree with everything in the above quote, except for the part about uncertainty estimates focusing on cooperation (which is a point raised by all reviewers). In fact the use of average target values, which is meant to "exhibit lower variance compared to commonly applied target networks, leading to significant benefits in MARL which commonly suffers from high variance caused by the exploration and non-stationary policies of other agents" seems to run somewhat against the main idea of uncertainty estimation (a point raised by reviewer ZqwK). More importantly, this seems to be mostly an empirical decision that improves performance, but is not central to uncertainty estimation for improved coordination.

During the discussion, the authors responded to concerns about the connection between uncertainty estimation and improved coordination by pointing to the diversity of environments and improved performance. While I appreciate the diversity of environments considered, this does _not_ prove that states with high value estimation uncertainty are always states where coordination is necessary. For this to be convincing, the authors should either provide theoretical evidence and/or empirical evaluations that go deeper than simply comparing performance curves.

Thus, although I think this work holds merit and could be of interest to the MARL community, I agree with all the reviewers that in its current for it is not yet ready for publication. Specifically, the authors should either substantially tone down the claims of "uncertainty in value estimates implies needed coordination", or provide stronger evidence for this claim.

**Resubmission Of Major Revision:**

The authors may consider submitting a major revision at a later time.